# STAFF: SPECULATIVE CORESET SELECTION FOR TASK-SPECIFIC FINE-TUNING

**Xiaoyu Zhang**[1]    **Juan Zhai**[2]    **Shiqing Ma**[2]    **Chao Shen**[1]*
**Tianlin Li**[3]    **Weipeng Jiang**[1]    **Yang Liu**[3]
[1]Xi'an Jiaotong University    [2]University of Massachusetts, Amherst
[3]Nanyang Technological University
{zxy0927@stu.xjtu,chaoshen@xjtu,lenijwp@stu.xjtu}.edu.cn
{juanzhai,shiqingma}@umass.edu
{tianlin001@e.ntu,yangliu@ntu}.edu.sg

## ABSTRACT

Task-specific fine-tuning is essential for the deployment of large language models (LLMs), but it requires significant computational resources and time. Existing solutions have proposed coreset selection methods to improve data efficiency and reduce model training overhead, but they still have limitations: ❶ Overlooking valuable samples at high pruning rates, which degrades the coreset's performance. ❷ Requiring high time overhead during coreset selection to fine-tune and evaluate the target LLM. In this paper, we introduce STAFF, a speculative coreset selection method. STAFF leverages a small model from the same family as the target LLM to efficiently estimate data scores and then verifies the scores on the target LLM to accurately identify and allocate more selection budget to important regions while maintaining coverage of easy regions. We evaluate STAFF on three LLMs and three downstream tasks and show that STAFF improves the performance of SOTA methods by up to 54.3% and reduces selection overhead by up to 70.5% at different pruning rates. Furthermore, we observe that the coreset selected by STAFF at low pruning rates (i.e., 20%) can even obtain better fine-tuning performance than the full dataset. Our code is publicly available at https://github.com/shiningrain/STAFF.

## 1 INTRODUCTION

Large Language Models (LLMs) have shown great potential in solving various problems such as text summarization, machine translation, code generation (Van Veen et al., 2024; Roziere et al., 2023; Gong et al., 2024; Zhao et al., 2024). To achieve the best performance on downstream tasks, fine-tuning these foundational models on task-specific datasets is necessary. This process, known as the task-specific LLM fine-tuning, demands high computational resources and time because of the large size of LLMs and datasets. It leads to high carbon emissions, hurting the economy and environment (Faiz et al., 2024). To reduce the training overhead and improve data efficiency, researchers have proposed various data pruning and coreset selection methods, mainly consisting of data importance-guided methods and data diversity-guided methods (Maharana et al., 2024). Data importance-guided methods leverage scoring functions to evaluate the importance or difficulty of samples based on their training record and retain the most difficult or important samples to construct the coreset for DL models (Paul et al., 2021; Toneva et al., 2018). Data diversity-guided methods first divide or cluster the samples into different regions and areas based on their scores and then select samples across data regions to ensure that the coreset can represent different regions (Zheng et al., 2023; Maharana et al., 2024).

However, existing coreset selection methods have limitations and are not good for task-specific LLM fine-tuning. On the one hand, existing methods have difficulty in balancing data importance and diversity in coreset selection. As a result, they could ignore representative samples in regions with low scores at medium-high pruning rates, leading to catastrophic performance degradation

---

*Chao Shen is the corresponding auther.

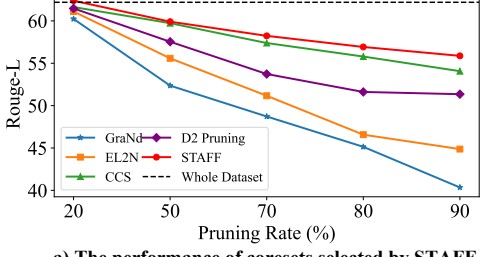

| Method | Rouge-L ↑ | Time (h) ↓ |
|--------|-----------|------------|
| D2 Pruning | 51.4 | 11.8 |
| CCS | 54.1 | 8.9 |
| STAFF | 55.9 | 3.5 |
| **Impr. (%)** | **2.2~8.8** | **61.0~70.5** |

a) The performance of coresets selected by STAFF and baseline methods at different pruning rates.

b) The comparison between STAFF and two optimal baselines on the coreset performance and time overhead (on NVIDIA RTX A6000) at 90% pruning rate.

**Figure 1:** Experiment results on the WMT-19 dataset and Gemma-7b model. **a)** reveals the effectiveness of STAFF in coreset selection across different pruning rates. **b)** shows the low selection overhead of STAFF.

of the selected coreset on the fine-tuned model (Zheng et al., 2023). Figure 1 a) provides a demo case that, when the pruning rate is increased from 20% to 90%, the performance of coreset (i.e., Rouge-L of the fine-tuned model) selected by the data importance-guided methods GraNd and EL2N separately decreases by 33.0% and 26.6%. On the other hand, existing methods require training the target model on the given dataset for several epochs to evaluate data scores and regions during coreset selection (Paul et al., 2021; Maharana et al., 2024). Such a training and evaluation process is acceptable on small-scale DL models but results in an extremely time-consuming selection process on LLMs. Our experiment shows that the selection process may take more than ten hours on powerful devices (Figure 1 b)). There is an urgent need for an effective and efficient coreset selection method that performs well across different pruning rates and has low selection overhead.

We observe that LLMs in the same family have similar model architectures and usually share the same datasets in pre-training (Touvron et al., 2023), leading to their similar knowledge distribution and similar feedback to a given input. These small models have been widely used to accelerate the decoding of target LLMs, and have shown strong capabilities in predicting LLM output tokens and accelerating inference (Miao et al., 2024; Leviathan et al., 2023). Building upon this observation, inspired by the concept of *speculative execution* in computer architecture, we propose STAFF, a speculative coreset selection method that employs a small model from the same family as the target LLM to efficiently evaluate the importance score of data. Subsequently, it verifies the score on the target model, accurately identifies the important data regions, and effectively selects the coreset.

Specifically, STAFF consists of two stages. ❶ **Speculative score calculation** first fine-tunes the small model and uses it to efficiently calculate the importance score (i.e., speculative score) of each sample in the downstream dataset. It uses the effort score function (Paul et al., 2021) by default to measure the parameter changes when the model learns and fits each sample, reflecting the difference between the distribution of downstream task and the pre-trained knowledge distribution of the model. ❷ **LLM Verification & Selection** then conducts stratified sampling to divide data regions based on the speculative scores, and verifies and evaluates the importance of different data regions for the target LLM. During the coreset selection, STAFF simultaneously covers both important (difficult) and easy regions, dynamically allocating more selection budget to the former to include important samples as much as possible. Such a design prioritizes the removal of easy samples at low pruning rates and ensures the selection of diverse data to represent different data regions at high pruning rates, thus balancing the data importance and diversity in the coreset selection across different pruning rates.

To verify the effectiveness and efficiency of STAFF in coreset selection for various downstream datasets, we compare STAFF with five state-of-the-art (SOTA) selection methods on three LLMs and three downstream tasks (i.e., biology question-answering, dialogue summarization, and translation of minority languages). Experiment results show that STAFF outperforms SOTA methods in coreset selection across different pruning rates, improving fine-tuning performance by up to 54.3% compared to the best baseline method and saving up to 70.5% of selection overhead. Additionally, we observe that the coreset selected by STAFF at low pruning rates (e.g., 20%) has the potential to achieve better fine-tuning results than the full dataset, further demonstrating the effectiveness of STAFF in selecting coreset and improving data efficiency. Figure 1 shows the comparison of STAFF and SOTA methods on the Gemma-7b model and WMT-19 dataset under different data pruning rates. It demonstrates that STAFF can achieve significantly better results than the existing methods at different pruning rates

and reduce the selection time overhead by over 60% compared with the two optimal baselines, which illustrates the effectiveness and efficiency of STAFF. In summary, our contributions are:

- We extend and apply the concept of *speculative execution* to the coreset selection to improve data efficiency of task-specific LLM fine-tuning, providing valuable insights for future work.
- We propose STAFF, a coreset selection method based on the concept of speculative execution that leverages the small model to efficiently guide the coreset selection for the target LLM.
- We evaluate STAFF on three LLMs and three downstream tasks, and STAFF outperforms SOTA methods across different pruning rates with a maximum reduction of 70.5% in selection overhead.
- Our implementation and data are publically available[1].

## 2 PRELIMINARIES

### 2.1 CORESET SELECTION FOR TASK-SPECIFIC FINE-TUNING

The pre-trained LLMs are fine-tuned with different settings to be harmless or generalize to unseen tasks, such as instructional fine-tuning (Liu et al., 2023; Muennighoff et al., 2023; Lu et al., 2023; Du et al., 2023; Chen et al., 2023) and preference fine-tuning (Ethayarajh et al., 2024; Pace et al., 2024; Kim et al., 2023; Cui et al., 2023). This paper focuses on task-specific fine-tuning that updates the model to adapt the target distribution of a given downstream task, which is essential for the deployment and application of LLMs (Yang et al., 2024; Lin et al., 2024; Albalak et al., 2024; Han et al., 2024; Wang et al., 2024a). One-shot coreset selection for task-specific fine-tuning aims to remove redundant data and improve data efficiency. Consider a downstream task dataset $\mathbb{D}$ containing $N$ samples $\mathbb{D} = \{(x_i, y_i)\}_{i=1}^N$ drawn i.i.d. from the distribution $\mathbb{P}$. Coreset selection methods select a subset $\mathbb{D}'$ at the given prune rate $p$ such that the loss of the LLM $\theta$ trained on $\mathbb{D}'$ using loss function $L$ is minimized on the test set drawn from $\mathbb{P}$. The optimization problem can be expressed as:

$$\min_{\mathbb{D}' \subseteq \mathbb{D}: \frac{|\mathbb{D}'|}{|\mathbb{D}|} \leq 1-p} \mathbb{E}_{x,y \sim \mathbb{P}} \left[ L(x, y; \theta(\mathbb{D}')) \right] \tag{1}$$

Prior work has proposed various coreset selection methods (Chai et al., 2023; Xia et al., 2022; Yang et al., 2024; Lin et al., 2024), mainly considering two factors of data importance and data diversity (Maharana et al., 2024). ❶ **Data Importance.** Researchers have proposed various statistical metrics (e.g., difficulty score and effort score (Guo et al., 2022; Paul et al., 2021)) to select important or difficult data in the dataset without degrading test performance. However, data with a low importance score also plays an important role in model training and optimization (Baldock et al., 2021; Paul et al., 2022). Zheng et al. (2023) have demonstrated the necessity of including low-score examples to cover high-density regions of the dataset, which leads to increased attention to data diversity in existing work. ❷ **Data Diversity.** Prior works have proposed various methods to assess data diversity and select coresets from different data regions to guarantee representation (Chan et al., 2022; Yu et al., 2022; Seki et al., 2024; Lee et al., 2024). Zheng et al. (2023) divide the dataset distribution into several areas based on the data importance score, and they observe that data with high scores is usually sparse and lies in low-density areas. Their selection method covers both high-density and low-density areas and achieves good results in experiments. Maharana et al. (2024) merge and filter out similar data samples based on message-passing algorithms and select representative samples from all regions of data distribution.

### 2.2 SPECULATIVE EXECUTION

Speculative execution is an optimization strategy in the field of computer architecture (Patterson et al., 2011) to improve the efficiency of upcoming tasks. It mainly consists of the execution stage and verification stage, as shown in Figure 2. Specifically, consider an upcoming task $T : x \mapsto Y$, where $x$ is the task input and $Y$ is the corresponding solution space. The execution stage utilizes limited spare resources to perform the speculative task $Spec(\cdot)$ and obtain a result $z$ for the given

---

[1] Our code is available at `https://github.com/shiningrain/STAFF`

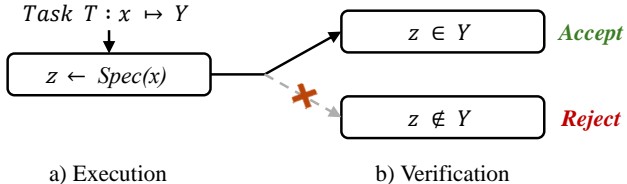

**Figure 2:** Speculative execution uses the speculative task $Spec(\cdot)$ to speed up the upcoming task $T$.

input $x$, as shown in Figure 2 a), Then, the verification stage verifies the result $z$ (Figure 2 b)). If $z$ is useful and can pass the verification (i.e., $z \in Y$), it will be accepted, otherwise, $z$ will be rejected. This process will not incur excessive additional costs.

The concept of *speculative execution* has been widely used in LLM to improve the quality and efficiency of decoding and reasoning (Miao et al., 2024; Leviathan et al., 2023; Hooper et al., 2023; Li et al., 2024b; Xia et al., 2024; Zhang et al., 2024b; Li et al., 2024a). Li et al. (2024b) uses the small model to purify the target LLM and mitigate issues such as copyright infringement and privacy violations while preserving the performance of the target LLM. Miao et al. (2024) propose SpecInfer that first uses the small speculative models to generate the token sequences and then proposes parallel decoding to verify the generated token sequences efficiently. In this process, task $T$ in Figure 2 is to generate token sequences $y \in Y$ for the model inputs $x$, while the speculative task $Spec(\cdot)$ is to use a small model to generate a token sequence $z$ based on the input $x$. If $z$ belongs to the target distribution $Y$, the token sequence $z$ will be accepted and used to generate new tokens.

In this paper, we extend and apply the concept of speculative execution to coreset selection for the task-specific fine-tuning of the target LLM, aiming to both improve the effectiveness of coreset selection and reduce the selection overhead. Existing LLMs typically have several variants with different parameter scales, forming a family, such as Llama-2-13b and Llama-2-7b (Touvron et al., 2023). In this scenario, the upcoming task $T$ is to calculate the importance score of each sample on the target LLM. We use a small model in the same family of the target LLM to evaluate the score of samples with much lower overhead (i.e., the speculative task $Spec(\cdot)$), then verify the results on the target LLM and select the coreset without the fine-tuning of the target LLM.

## 3 METHODOLOGY

Although existing data importance-guided and diversity-guided coreset selection methods have demonstrated good performance in scenarios such as deep learning (DL) training, they still face two major challenges in task-specific LLM fine-tuning. ❶ **Challenge 1: Effective coreset selection across all pruning rates.** Existing selection methods can hardly balance data importance and diversity in selection, especially at higher pruning rates. As the pruning rate increases, they often ignore representative samples from regions with low importance scores, leading to catastrophic performance degradation, which is also observed in DL model training (Zheng et al., 2023). Figure 1 a) shows an intuitive case where existing selection methods can lead to performance degradation of up to 35.1% in the fine-tuned target LLM when the pruning rate increases from 20% to 90%. Ensuring the effective selection of diverse and important samples at both high and low pruning rates remains a significant challenge. ❷ **Challenge 2: Reducing overhead in coreset selection.** Existing methods typically involve high time overhead for LLMs, as they require several epochs of training on the target model to evaluate data scores and regions during selection. While this process might be manageable for light-weight DL models, it becomes prohibitively expensive for LLM fine-tuning tasks, where large datasets and massive model parameters are involved. This can lead to a selection process taking dozens of GPU hours (Figure 1 b)), limiting the efficiency and applicability of these selection methods. Designing an efficient method to evaluate and select samples with reduced time overhead is a critical challenge.

To address the aforementioned challenges, we propose STAFF, a speculative coreset selection method that introduces a small model in the same family as the target LLM to evaluate the data scores and then uses the target model for validation and selection. STAFF does not require heavy fine-tuning of the target model in selection, achieving effective coreset selection with low overhead at different pruning rates. STAFF consists of two stages, and Figure 3 shows the workflow. In the *speculative*

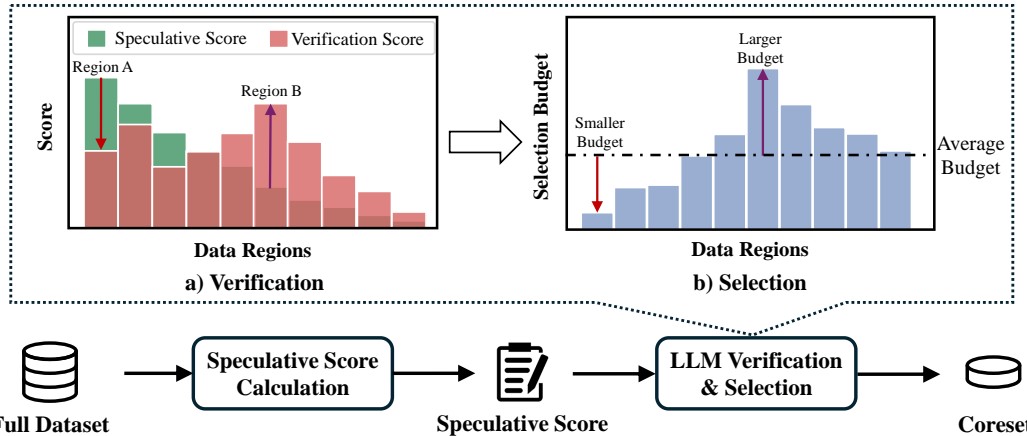

**Figure 3:** The Overview of STAFF. In LLM verification and selection, STAFF **a)** verifies the score of different data regions on the target LLM and then **b)** adjusts the selection budget based on the difference between the speculative score and the verification score on the target LLM (e.g., 'Region A&B') to cover data regions that are important to the target LLM.

*score calculation* stage, STAFF fine-tunes the small model and evaluates the data importance score on the fine-tuned small model (green bars in the dashed box). Then, in the *LLM verification & selection* stage, based on the speculative scores and verification scores (red bars), it accurately identifies the regions important to the target LLM and adjusts the selection budget allocated to each region to select more important samples while ensuring data diversity. The algorithm of STAFF is represented in Algorithm 1, where the *speculative score calculation* stage is shown in Line 2 to 3 and the *LLM verification & selection* stage is shown in Line 4 to 13.

## 3.1 SPECULATIVE SCORE CALCULATION

To find the important samples for the target model $\theta_t$ on the dataset $\mathbb{D}$, the most intuitive approach is to train $\theta_t$ on $\mathbb{D}$ for a few epochs and then let $\theta_t$ itself to evaluate the samples based on the training performance (Maharana et al., 2024; Zheng et al., 2023). However, training LLMs is very time-consuming. We have observed that models in the same family have similar architecture and are usually pre-trained on the same corpus (Touvron et al., 2023). As a result, they encode similar knowledge in their parameters. Due to these similarities, models in the same family exhibit similar evaluation results and scores for data samples (Figure 4), while the overhead of small models is only half or even less than those of the large ones.

Based on such observation, in this stage, we use the small model $\theta_s$ in the same family as $\theta_t$ to perform 'speculative execution' with much lower selection overhead, which is fine-tuning $\theta_s$ on dataset $\mathbb{D}$ to calculate data scores. We have further studied the impact of using small models from other families in §4.3. After fine-tuning $\theta_s$, we introduce the effort score of a sample $d$ (Paul et al., 2021; Lin et al., 2024) to calculate the speculative score, as follows:

$$S_d^s = \|\nabla_\phi L(\theta_s(d))\|_2 \,, \tag{2}$$

where $\phi$ indicates the learnable parameters in the small model $\theta_s$. The score set in Line 3 has $S^s = \{S_d^s, \forall d \in \mathbb{D}\}$. The effort score measures the change in model parameters when learning the given sample $d$ and reflects the learning effort of $\theta_s$ to fit the given input. A larger effort score indicates a greater difference between the sample and the knowledge currently encoded in the pre-trained model, suggesting that it is more difficult for the model to learn and fit this sample. Considering the small model $\theta_s$ has a similar architecture and pre-trained corpus to the target LLM $\theta_t$, the effort score calculated on $\theta_s$ has the potential to reflect the importance and difficulty of the data to $\theta_t$. As a result, we use the effort score to perform the speculative task on $\theta_s$. §A.4 study the impact of using other scoring functions (e.g., influence score) from existing work (Koh & Liang, 2017).

---

**Algorithm 1** STAFF for Coreset Selection

---

**Input:**    $\mathbb{D}$ — the full dataset ($\{(x_i, y_i)\}_{i=1}^N$);        $\theta_t$ — the target LLM (w/o finetuning);
           $\theta_s$ — the small LLM (w/o finetuning) in the same family as the target LLM;    $p$ — the pruning rate;
           $T$ — the maximun fine-tuning budget for small LLM;            $K$ — the number of regions;
           $b_v$ — the budget of LLM verification;

**Output:**   $\mathbb{D}'$ — the selected coreset ($\frac{|\mathbb{D}'|}{|\mathbb{D}|} \leq 1 - p$);

1: **procedure** CORESETSELECTION($\mathbb{D}, \theta_t, \theta_s, p, T, K, b_v$)
2:     $\theta_s \leftarrow fineTuning(\theta_s, \mathbb{D}, T)$                                               ▷ Fine-tune small model
3:     $S^s \leftarrow getScore(\theta_s, \mathbb{D})$                         ▷ Calculate speculative score $S^s$ on the full dataset
4:     $\mathcal{B} = \{B_1, B_2, ... B_K\}$           ▷ Split $\mathbb{D}$ into $K$ regions according to $S^s$ with even range width
5:     $m \leftarrow N \times (1 - p)$                           ▷ Initial total budget $m$ in coreset selection
6:     $\mathbb{D}' \leftarrow \varnothing; k \leftarrow 0$
7:     **while** $\mathcal{B} \neq \varnothing$ **do**                                  ▷ Select samples from all regions in $\mathcal{B}$
8:        $B_i \leftarrow \arg \min_{B_i \in \mathcal{B}} |B_i|$
9:        $B_i^* \leftarrow$ randomly select $\min\{b_v, |B_i|\}$ samples from $B_i$
10:      $\mathcal{V}_i \leftarrow verify(B_i^*, \theta_t, S^s)$                      ▷ Get verification results of each region
11:      $m_B \leftarrow \lfloor \frac{(m - |\mathbb{D}'|)\mathcal{V}_i}{|\mathcal{B}|} \rfloor$        ▷ Calculate selection budget based on verification result $\mathcal{V}_i$ of $B_i$
12:      $\mathbb{D}_i \leftarrow$ randomly select $\min\{m_B, |B_i|\}$ samples from $B_i$
13:      $\mathbb{D}' \leftarrow \mathbb{D}' \cup \mathbb{D}_i; \mathcal{B} \leftarrow \mathcal{B} \setminus B_i$                         ▷ Update coreset $\mathbb{D}'$
14:     **return** $\mathbb{D}'$

---

## 3.2 LLM VERIFICATION & SELECTION

In this stage, we use the target LLM $\theta_t$ to verify the speculative score $S^s$ from the small model $\theta_s$ (Line 4 to 10) and then select coreset $\mathbb{D}'$ based on the verification results (Line 11 to 13). Our goal is to retain those important data for the target LLM $\theta_t$ while covering different data regions to ensure diversity and enable the coreset to achieve good performance at both high and low pruning rates.

**Verification.** We first follow the stratified sampling in Zheng et al. (2023) to divide the dataset $\mathbb{D}$ into $K$ regions with equal range width based on the speculative score $S$ (Line 4). This stratified division ensures that the subsequent validation and selection cover diverse data regions with varying importance scores, thereby improving the diversity of the coreset. Then, in the verification and selection, we prioritize data regions with fewer samples, as sparse regions usually contain important and difficult samples and dense regions contain easy examples (Zheng et al., 2023) (Line 8). We randomly select $b_v$ samples from the target region $B_i$ to verify them on the target model $\theta_t$ and calculate the verification result $\mathcal{V}_i$ in Line 10, as shown in follows:

$$\mathcal{V}_i = \frac{\sum_{d \in B_i^*} S_d^t}{\sum_{d \in B_i^*} S_d^s}, \tag{3}$$

where $S_d^s$ indicates the speculative score of sample $d$ calculated on $\theta_s$ and $S_d^t$ is the verification score on $\theta_t$ (also uses the effort score). The score $\mathcal{V}_i$ reflects the difference in the data importance in the region $B_i$ for the target LLM $\theta_t$ and the small model $\theta_s$. $\mathcal{V}_i > 1$ indicates that the data in this region is more important to the target LLM $\theta_t$ (e.g., 'Region B' in Figure 3 a)), and selecting data from this region can help $\theta_t$ effectively modify model parameters and learn the downstream task of $\mathbb{D}$. A small $\mathcal{V}_i$ indicates that the data in $B_i$ is not so important to the target model $\theta_t$ (e.g., 'Region A').

**Selection.** We select data from each region based on the verification result $\mathcal{V}_i$ and allocate more selection budget to those data regions that are important to $\theta_t$ and reduce the budget for easy regions, thus achieving a coreset selection that balances data importance and diversity (Line 11). For the selected $B_i$, we use its verification result $\mathcal{V}_i$ to calculate the selection budget $m_B$:

$$m_B = \lfloor \frac{(m - |\mathbb{D}'|)\mathcal{V}_i}{|\mathcal{B}|} \rfloor, \tag{4}$$

where $m$ is the total selection budget for the coreset $\mathbb{D}'$ and $\frac{(m - |\mathbb{D}'|)}{|\mathcal{B}|}$ represents the average selection budget of each region that has not been selected. $\mathcal{V}_i$ is the correction factor of the selection budget, aiming to allocate more budget to those important data regions. Take the data region marked by 'Region B' in Figure 3 a) as an example. The verification result $\mathcal{V}_i$ in this region is much larger than 1, indicating that the samples in this region are much more important to the target model than to the small model. To retain important samples in the coreset, STAFF allocates a larger selection budget to the data in this region during selection, as shown in the blue bars in Figure 3 b). To ensure the data

diversity in the coreset, STAFF also selects samples from less important regions (e.g., 'Region A'), but with a smaller selection budget. This design ensures that, at low pruning rates, STAFF can exclude easy samples that are not so important to the target model $\theta_t$ and focus on the important samples, while still guaranteeing coverage of different data regions at high pruning rates. Finally, we randomly select samples from $B_i$ based on the budget $m_b$, merge them with the coreset set $\mathbb{D}'$ and update $\mathcal{B}$.

Through these stages, STAFF can accurately identify data regions that are important to the target LLM with a much smaller selection overhead. It considers both data importance and diversity in selection and allocates a higher selection budget to important and difficult regions and a lower budget to easy ones. Ultimately, it outperforms baseline methods on multiple downstream tasks and LLMs (§4.2).

## 4 EXPERIMENT

### 4.1 EXPERIMENT SETUP

**Tasks, Models & Datasets.** We evaluate STAFF on three datasets on different downstream tasks, namely, the **BioInstruct** dataset (Tran et al., 2024) (biology question-answering), **DialogSum** dataset (Chen et al., 2021) (dialogue summarization), and the 'Kazakh-English' subset of **WMT-19** dataset (Barrault et al., 2019) (translation of minority languages). We fine-tune and evaluate three popular LLMs in the experiments on these downstream tasks, namely **Gemma-7b** (Team et al., 2024), **Llama-2-13b** (Touvron et al., 2023), and **Mistral-Nemo-Instruct-2407** (Jiang et al., 2023). We use Gemma-2b, Llama-2-7b, and Mistral-7B-Instruct-v0.2 as the small model of the corresponding LLMs in the same family. All models can achieve significant performance improvement after fine-tuning.

**Baselines.** We compare STAFF with several state-of-the-art coreset selection methods. ❶ **Random** selection obtain the subset via random sampling. ❷ **GraNd** (Paul et al., 2021) selects the difficult samples with larger gradient norms to construct the coreset. ❸ **EL2N** (Paul et al., 2021; Marion et al., 2023) selects the difficult samples whose prediction results are more different from the ground truth sequences. ❹ **CCS** (Zheng et al., 2023) constructs coresets considering both data coverage and importance. We use EL2N as its important metric. ❺ **D2 Pruning** (Maharana et al., 2024) constructs graphs to update data scores and selects samples from diverse regions.

**Implementations.** All fine-tuning experiments are conducted on one NVIDIA RTX A6000 GPU. Besides, we use the parameter-efficient technique 'LoRA' (Hu et al., 2022) in fine-tuning. The number of samples used in verification for each bin ($b_v$) is 10. Based on the results of prior work (Paul et al., 2021; Zheng et al., 2023), we set fine-tuning budget $T$ in selection to 3 and $K$ to 50. More hyperparameter settings are in §A.3. We use Rouge-L as the metric to evaluate the model fine-tuned on the coresets in experiments. In addition, we count the total time overhead during the coreset selection process of each selection method except 'Random', including fine-tuning the target model to calculate importance scores, merging similar samples, and selecting the coreset.

### 4.2 COMPARISON WITH BASELINES

We evaluate STAFF and baseline methods on three LLMs and three downstream tasks across different pruning rates (20% to 90%), and fine-tuning results are presented in Table 1. We can observe that STAFF can perform better than state-of-the-art methods at both low and high pruning rates.

**Comparison among different methods.** The data importance-guided methods (e.g., EL2N (Paul et al., 2021)) can achieve competitive results at low pruning rates. For example, using EL2N on the Mistral-Nemo model and the BioInstruct dataset can achieve the best results among all methods at $p = 20\%$. However, these methods suffer from catastrophic performance degradation at high pruning rates, especially on the WMT-19 dataset. Using the GraNd method on the Llama-2-13b model and WMT-19 dataset experiences a 39.4% performance degradation as $p$ increases from 20% to 90%, which echoes the observation in Figure 1. In contrast, STAFF can achieve good results at different pruning rates. For example, on the WMT-19 dataset, STAFF outperforms the best baseline method by 3.3%, 2.2%, and 1.9% at a high pruning rate ($p = 90$). Moreover, we observe that the coresets selected by STAFF experience little performance degradation at low pruning rates and can even achieve better fine-tuning results than the full dataset, which is not observed in baseline methods. For instance, on the DialogSum dataset with $p = 20\%$, STAFF exhibits an average improvement

**Table 1:** Performance (Rouge-L) of STAFF and baselines on three datasetsl; across different pruning rates (20% to 90%). (Higher is better. Bold indicates the best result, italic and underlined indicate the second-best result.)

| Model | Method | BioInstruct | | | | | | DialogSum | | | | | | WMT-19 | | | | | |
|---|---|---|---|---|---|---|---|---|---|---|---|---|---|---|---|---|---|---|---|
| | | 0% | 20% | 50% | 70% | 80% | 90% | 0% | 20% | 50% | 70% | 80% | 90% | 0% | 20% | 50% | 70% | 80% | 90% |
| Gemma-7b | Random | 41.8 | 41.1 | 40.6 | 39.5 | 38.8 | 38.5 | 49.8 | 48.0 | 47.0 | 47.0 | 46.8 | 45.5 | 62.2 | 61.0 | 58.8 | 57.0 | 55.7 | 53.7 |
| | GraNd | - | 40.3 | 39.7 | 39.1 | 37.9 | 37.8 | - | 48.7 | 47.8 | 46.8 | 46.6 | 45.5 | - | 60.2 | 52.4 | 48.7 | 45.1 | 40.3 |
| | EL2N | - | 41.3 | 41.3 | 40.4 | 39.8 | 38.8 | - | 49.5 | 48.6 | 47.7 | 47.0 | 46.2 | - | 61.1 | 55.6 | 51.2 | 46.6 | 44.9 |
| | CCS | - | 41.7 | 41.5 | 40.3 | 40.6 | 39.3 | - | 49.6 | 48.1 | 48.3 | 48.2 | 47.4 | - | 61.6 | 59.7 | 57.4 | 55.8 | 54.1 |
| | D2 Pruning | - | 41.4 | 41.2 | 40.5 | 40.2 | 39.1 | - | 49.4 | 48.8 | 47.6 | 46.9 | 45.9 | - | 61.5 | 57.5 | 53.7 | 51.6 | 51.4 |
| | STAFF | - | 42.0 | 41.5 | 41.3 | 40.8 | 39.4 | - | 50.0 | 48.9 | 48.9 | 48.5 | 48.4 | - | 62.4 | 59.9 | 58.2 | 56.9 | 55.9 |
| Llama-2-13b | Random | 41.7 | 41.0 | 40.3 | 39.5 | 38.6 | 37.9 | 49.6 | 49.1 | 48.1 | 46.9 | 46.2 | 45.5 | 62.1 | 60.6 | 58.4 | 55.8 | 54.5 | 52.1 |
| | GraNd | - | 40.5 | 39.1 | 37.8 | 36.9 | 35.2 | - | 49.3 | 48.2 | 46.6 | 45.4 | 44.7 | - | 58.7 | 51.8 | 44.8 | 40.3 | 35.5 |
| | EL2N | - | 40.7 | 39.8 | 39.6 | 38.9 | 38.5 | - | 48.9 | 48.5 | 47.7 | 47.3 | 45.9 | - | 60.0 | 55.1 | 50.6 | 48.3 | 42.8 |
| | CCS | - | 41.7 | 41.0 | 40.5 | 39.9 | 38.5 | - | 48.9 | 49.1 | 48.8 | 48.0 | 47.5 | - | 61.2 | 59.3 | 57.2 | 56.0 | 53.6 |
| | D2 Pruning | - | 41.4 | 40.7 | 40.2 | 39.7 | 38.4 | - | 48.8 | 48.4 | 48.0 | 47.0 | 46.1 | - | 61.4 | 58.7 | 55.8 | 52.4 | 49.4 |
| | STAFF | - | 42.2 | 41.6 | 40.7 | 40.2 | 39.8 | - | 50.0 | 49.1 | 49.0 | 48.4 | 48.2 | - | 62.0 | 59.5 | 56.9 | 56.2 | 54.8 |
| Mistral-Nemo | Random | 43.2 | 42.8 | 41.3 | 40.6 | 40.5 | 40.2 | 50.3 | 49.7 | 48.3 | 48.2 | 47.1 | 47.1 | 65.8 | 64.8 | 62.5 | 61.1 | 59.8 | 58.4 |
| | GraNd | - | 41.5 | 40.2 | 39.5 | 38.6 | 37.8 | - | 49.6 | 49.2 | 47.9 | 47.5 | 45.9 | - | 64.4 | 59.4 | 53.2 | 50.5 | 44.9 |
| | EL2N | - | 43.2 | 42.3 | 41.4 | 41.2 | 40.2 | - | 50.3 | 48.4 | 47.8 | 46.9 | 46.4 | - | 64.7 | 61.5 | 58.3 | 56.0 | 51.9 |
| | CCS | - | 43.0 | 42.5 | 42.2 | 41.9 | 40.6 | - | 50.7 | 49.0 | 49.5 | 49.0 | 48.7 | - | 65.1 | 63.5 | 61.9 | 61.3 | 59.2 |
| | D2 Pruning | - | 42.9 | 42.7 | 41.2 | 41.0 | 40.0 | - | 50.2 | 48.8 | 47.6 | 47.1 | 46.3 | - | 65.0 | 61.9 | 59.1 | 56.8 | 53.6 |
| | STAFF | - | 43.1 | 42.3 | 42.2 | 41.2 | 41.2 | - | 50.4 | 50.2 | 49.2 | 49.2 | 48.9 | - | 65.4 | 64.1 | 63.1 | 62.1 | 60.3 |

**Table 2:** Time overhead (h) of coreset selection under $p = 90\%$ of STAFF and baselines on three datasets. (Lower is better. Bold indicates the best result, italic and underlined indicate the second-best result.)

| Method | Gemma-7b | | | Llama-2-13b | | | Mistral-Nemo | | |
|---|---|---|---|---|---|---|---|---|---|
| | Bioinstruct | DialogSum | WMT-19 | Bioinstruct | DialogSum | WMT-19 | Bioinstruct | DialogSum | WMT-19 |
| GraNd | 5.1 | 4.4 | 8.7 | 6.0 | 4.9 | 14.0 | 6.1 | 6.8 | 10.9 |
| EL2N | 5.4 | 4.8 | 8.9 | 6.5 | 5.5 | 15.4 | 6.5 | 7.2 | 11.9 |
| CCS | 5.4 | 4.8 | 8.9 | 6.5 | 5.5 | 15.4 | 6.5 | 7.2 | 11.9 |
| D2 Pruning | 7.7 | 5.4 | 11.8 | 7.7 | 6.8 | 20.5 | 7.4 | 8.3 | 15.1 |
| STAFF | 3.1 | 3.0 | 3.5 | 4.9 | 3.2 | 8.7 | 4.0 | 5.4 | 9.3 |

of 0.2 in fine-tuning results compared to the full dataset, further demonstrating the effectiveness of STAFF in pruning ineffective and redundant data and improving data efficiency.

While improving the coreset performance, STAFF also has a smaller time overhead than baseline methods in coreset selection. Table 2 shows the selection overhead of different methods under $p = 90\%$. We can observe that the baseline method D2 pruning incurs the largest time overhead in selection. It not only fine-tunes the target model to calculate the data scores but also leverages the message-passing algorithm to update scores and merge samples, which could take 20 hours on large datasets (e.g., WMT-19). In contrast, STAFF effectively reduces the selection overhead by 14.7% to 70.5% compared to baseline methods on three datasets. Using smaller models in selection can lead to more significant efficiency improvements. STAFF has achieved the most significant improvement in selection efficiency when using the Gemma-2b model to calculate speculative scores for the Gemma-7b model. Taking the WMT-19 dataset as an example, STAFF improves the performance of the baseline method by 2.2% to 38.5% while reducing the selection time by 59.8% to 70.5%.

**Comparison among different LLM.** We observe that all methods experience the most significant performance degradation on the Llama-2-13b model as the pruning rate increases. Using GraNd separately results in performance degradation of 13.2%, 9.4%, and 39.4% on three datasets. Moreover, coreset selection methods generally achieve better performance on the Mistral-Nemo model. For example, fine-tuning the Mistral-Nemo model on only 30% of the BioInstruct dataset selected by STAFF (42.2) still outperforms fine-tuning Gemme-7b and Llama-2-13b (41.8 and 41.7) on the full dataset. This performance difference may be related to the LLM's architecture and pre-training corpus. Different from the other two LLMs using the 'SentencePiece' tokenizer, Mistral-Nemo employs a new tokenizer 'Tekken', which can effectively and efficiently compress and handle various languages and source code. Additionally, Mistral-Nemo has been pre-trained on a larger scale corpus and has achieved better performance on various benchmarks compared to the other two models (Jiang et al., 2023), which might lead to better performance on downstream tasks.

**Table 3:** Ablation study results on Gemma-7b. Bold marks the best results (Rouge-L).

| Method | BioInstruct | | | | | DialogSum | | | | | WMT | | | | |
|---|---|---|---|---|---|---|---|---|---|---|---|---|---|---|---|
| | 20% | 50% | 70% | 80% | 90% | 20% | 50% | 70% | 80% | 90% | 20% | 50% | 70% | 80% | 90% |
| STAFF | **42.0** | **41.5** | **41.3** | **40.8** | 39.4 | **50.0** | **48.9** | **48.9** | **48.5** | **48.4** | 62.4 | **59.9** | 58.2 | **56.9** | **55.9** |
| - w/o verification | 41.9 | 41.4 | 41.1 | 40.4 | 39.1 | 49.4 | 48.3 | 48.8 | 47.9 | 48.1 | **62.9** | 59.4 | 58.2 | 56.5 | 54.0 |
| - w/o small model | 41.0 | 40.3 | 40.2 | 39.3 | **39.7** | 49.3 | 48.6 | 48.9 | 48.0 | 47.9 | 61.5 | 59.8 | 58.2 | 56.4 | 55.4 |
| - other small model | 40.8 | 39.5 | 38.6 | 38.5 | 38.4 | 49.6 | 48.1 | 48.2 | 47.6 | 47.0 | 61.0 | 59.3 | 57.6 | 56.3 | 54.8 |

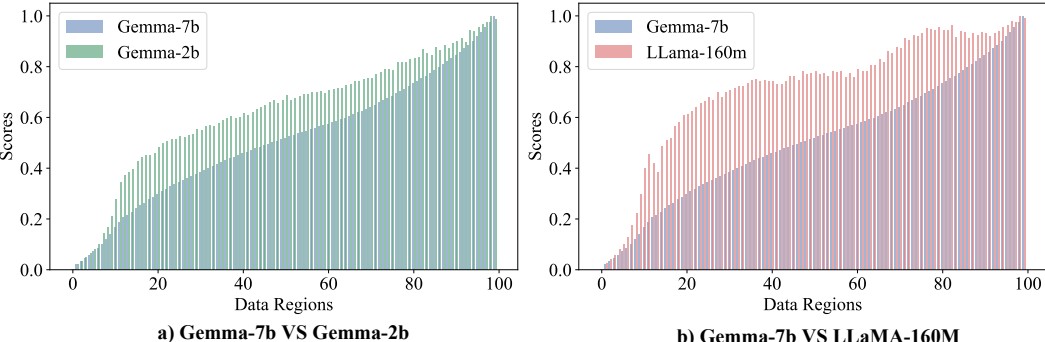

**a) Gemma-7b VS Gemma-2b**  **b) Gemma-7b VS LLaMA-160M**

**Figure 4:** The data score distribution of different models on the WMT-19 dataset. **a)** The data scores are highly similar across models in the same family (e.g., Gemma-7b and Gemma-2b). **b)** There are significant differences in score distributions across models from different families (e.g., Gemma-7b and LLama-160M (Miao et al., 2024)).

## 4.3 ABLATION STUDY

STAFF implements two stages to select coreset, namely 1) calculating the speculative score $S$ based on a small model that is in the same family as the target LLM and 2) verifying the score on the target LLM and calculating the selection budget $m_B$ based on the verification results $V_i$. To study the impact of these two stages on the selection results, we conduct an ablation study on the Gemma-7b model and present the results (Rouge-L) at different pruning rates in Table 3. Row 'w/o verification' shows the results without LLM verification, where the selection budget is evenly distributed to each region (which can be considered as $V_i$ always be 1). Row 'w/o small model' shows the results without speculative score calculation, where the selection directly utilizes the data score $S_d^t$ to divide regions $\mathcal{B}$, and then perform stratified sampling (Zheng et al., 2023) for each region. Row 'other small model' presents the results of using the speculative score calculated by a LLaMA-like small model (i.e., not in the Gemma family) with 160M parameters from Miao et al. (2024) (hereafter referred to as 'LLaMA-160M'). LLaMA-160M is pre-trained on a small corpus (e.g., part of the C4-en datasets) that is different from Gemma-7b's pre-training dataset.

We can observe that removing or replacing either stage leads to performance degradation, indicating the effectiveness and contribution of both stages within STAFF to the coreset selection. Specifically, 'other small model' achieves the most significant performance degradation. Our analysis indicates that due to the significant difference in training corpora and knowledge embedded in the model parameters, the speculative score calculated by this small model cannot effectively guide the data selection for Gemma-7b as a speculative model. Figure 4 exhibits the difference of score distribution among Gemma-7b, Gemma-2b, and LLaMA-160M models. We can observe that the data score distributions of the Gemma-7b and Gemma-2b models are very similar, which gradually increase from left to right, with slight differences in a few regions. In contrast, the data scores distribution of the LLaMA-160M model are significantly different, with scores wavy from left to right. As a result, substituting the Gemma-2b model with it leads to a significant performance degradation of STAFF across all pruning rates, and the fine-tuning results are close to those obtained by random selection. In addition, removing the verification on the target LLM ('w/o verification') also leads to performance degradation, especially on medium-high pruning rates. For example, at a pruning rate of 20%, 'w/o verification' can achieve performance close to or even better than STAFF. However, as the pruning rate increases, especially when the pruning rate is 90%, 'w/o verification' exhibits a performance degradation of up to 3.4% compared with STAFF. This is because, without LLM validation, the selection based solely on the speculative score from a small model cannot effectively

find important data regions for the target LLM, leading to performance degradation. Furthermore, directly using the data score $S_d^t$ from $\theta_t$ (i.e., 'w/o small model') to perform stratified sampling also leads to performance degradation. The untrained model $\theta_t$ performs poorly on the given downstream task and cannot accurately evaluate the importance score for the samples in the dataset.

## 5 RELATED WORK

**Coreset Selection.** Coreset selection has been widely used in machine learning and deep learning (Chai et al., 2023; Sener & Savarese, 2018; Xia et al., 2022). In addition to the aforementioned work, researchers have proposed various methods to select important, difficult, or diverse data to improve data efficiency (Xia et al., 2022; Coleman et al., 2019; Pleiss et al., 2020). However, these methods are usually designed for classification tasks and are susceptible to catastrophic performance degradation at high pruning rates. In addition, existing work also utilizes optimization techniques to optimize the data coreset, thereby minimizing empirical risks (Pruthi et al., 2020; Yang et al., 2023; Killamsetty et al., 2021). However, such complex optimization methods require cumbersome calculations and can hardly be applied to task-specific LLM fine-tuning. In this paper, we utilize small models to select the coreset taking account of both data diversity and importance, and achieve better-performing data selection with lower overhead than state-of-the-art methods.

**LLM Data Pruning.** Researchers have designed various data selection and pruning methods to improve the data quality and efficiency for LLM training, covering different settings like pre-training (Xie et al., 2023; Marion et al., 2023; Xie et al., 2023; Penedo et al., 2023), instructional fine-tuning (Liu et al., 2023; Muennighoff et al., 2023; Lu et al., 2023; Wang et al., 2022; Yang et al., 2024), and preference fine-tuning (Ethayarajh et al., 2024; Pace et al., 2024; Kim et al., 2023; Cui et al., 2023). This paper focuses on coreset selection for task-specific fine-tuning on various downstream tasks. Lin et al. (2024) have proposed a data selection method for the recommendation task of LLM systems based on the influence score and effort score. However, their method can hardly adapt to other tasks.

**Speculative Execution in LLM.** The concept of *speculative execution* has been widely used in LLM to improve the quality and efficiency of decoding and reasoning. (Miao et al., 2024; Leviathan et al., 2023; Hooper et al., 2023; Li et al., 2024b; Xia et al., 2024; Zhang et al., 2024b; Li et al., 2024a). Miao et al. (2024) propose SpecInfer that first uses the small speculative models to generate the token tree and then proposes multi-step speculative sampling and parallel decoding to efficiently verify the token tree. Inspired by the concept of *speculative execution*, in this paper, we use small models to calculate data score and perform verification and coreset selection on the target LLM, thereby achieving better results than existing methods while significantly reducing the selection overhead.

## 6 CONCLUSION

This paper introduces STAFF, a novel and efficient coreset selection method for task-specific LLM fine-tuning. STAFF first leverages a small model in the same family of the target LLM to calculate the data importance score. It then verifies the scores on the target LLM to accurately identify important regions and allocate larger selection budgets for them while ensuring the coverage of diverse data regions in selection. Our experiment results on three LLMs and three downstream tasks demonstrate that STAFF can effectively select the coreset at different pruning rates, improving the performance of SOTA methods by up to 54.3% while reducing selection time overhead by up to 70.5%.

## ACKNOWLEDGEMENTS

The authors thank the anonymous reviewers for their insightful feedback and constructive comments. Authors from Xi'an Jiaotong University are supported partially by the National Key Research and Development Program of China (2023YFB3107400), the National Natural Science Foundation of China (U24B20185, T2442014, 62161160337, 62132011, U21B2018), the Shaanxi Province Key Industry Innovation Program (2023-ZDLGY-38, 2021ZDLGY01-02). Thanks to the New Cornerstone Science Foundation and the Xplorer Prize. This research is supported by the National Research Foundation, Singapore, the Cyber Security Agency under its National Cybersecurity R&D Programme (NCRP25-P04-TAICeN), and DSO National Laboratories under the AI Singapore Programme (AISG2-GC-2023-008). It is also supported by the National Research Foundation, Prime Minister's Office, Singapore under the Campus for Research Excellence and Technological Enterprise (CREATE) programme.

## REPRODUCIBILITY STATEMENT

To follow the Open Science Policy and support reproducibility, we have released code about our implementations and evaluations. All resources are available in https://github.com/shiningrain/STAFF.

## ETHICS STATEMENT

This paper proposes an efficient and effective coreset selection method for task-specific LLM fine-tuning that does not involve potential violations of the Code of Ethics. Authors acknowledge that they have read and adhere to the Code of Ethics.

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

# A Appendix

The appendix is organized as follows:

**§A.1** provides the details of the datsets in our experiments.

**§A.2** introduces the baseline methods.

**§A.3** explains the selection of hyperparameters of STAFF.

**§A.4** provides additional experiment results (e.g., standard deviations of multiple runs).

**§A.5** discusses the potential enhancement of STAFF.

## A.1 Datasets

We use three datasets on three downstream tasks, namely, the **BioInstruct** dataset (Tran et al., 2024) (biology question-answering), **DialogSum** dataset (Chen et al., 2021) (dialogue summarization), and the 'Kazakh-English' subset of **WMT-19** dataset (Barrault et al., 2019) (translation of minority languages). The BioInstruct dataset contains 25,005 pairs of instruction and demonstration from a diverse range of biomedical and clinical NLP tasks, covering areas such as answering biomedical questions, assessing eligibility for clinical trials, etc. In this task, we use instructions as model input and fine-tune the target LLMs to generate the corresponding demonstration. The DialogSum dataset is a large-scale dialogue summarization dataset, consisting of 13,460 dialogues collected from public dialogue corpora, covering daily-life topics like schooling, work, medication, shopping, leisure, travel, etc. Each dialogue in the dataset has a corresponding summary as the ground truth. In this task, we take the dialogue as input and fine-tune the target LLMs to produce a corresponding summary. For the machine translation task, we use the 'Kazakh-English' subset in WMT-19 which is a new translation task in WMT-19. This subset contains 128,649 pairs of Kazakh and English sentences. In this task, we use the former as input and query the fine-tuned LLMs to generate the corresponding English sentences. In the experiment, we divided each dataset into the training set and the test set according to a ratio of 9:1.

## A.2 Baselines

Some coreset selection methods are designed for classification tasks and require classification labels to evaluate prediction results and select data (Toneva et al., 2018; Pleiss et al., 2020; Xia et al., 2022). They are difficult to apply to the above three downstream tasks. In this paper, we have collected 5 representative SOTA methods and compared them with STAFF. ❶ **Random.** It leverages random sampling to construct the coreset without any guidance. However, it may miss the important samples in the dataset at high pruning rates. ❷ **GraNd.** (Paul et al., 2021) This method uses the gradient norms of one sample on the given model as the data importance score and then selects the most important samples to construct coresets. ❸ **EL2N.** (Paul et al., 2021; Marion et al., 2023) It selects the difficult samples whose prediction results are more different from the ground truth sequences. ❹ **CCS** (Zheng et al., 2023). The Coverage-centric Coreset Selection (CCS) uses EL2N to score samples in the dataset and then divides the dataset into multiple regions based on the scores. It allocates the same budget to each region in coreset selection to ensure data diversity. ❺ $\mathbb{D}^2$ **Pruning** (Maharana et al., 2024) constructs graphs and uses the message passing algorithm to update data scores and merge the similar samples. It then selects representative samples from different data regions.

All hyperparameters of baseline methods follow the recommendations in their paper and open-source implementations. Based on the experiment results of Paul et al. (2021), each baseline method and STAFF train the model for 3 epochs (i.e., $T = 3$) when calculating data scores.

## A.3 Hyperparameters

**Coreset Selection.** We use the recommended hyperparameters in Zheng et al. (2023) to perform stratified sampling in verification and selection (i.e., the number of regions $K = 50$). Based on the experiment results of prior work Paul et al. (2021), we fine-tune models for three epochs to calculate and evaluate the data scores (i.e., $T = 3$). We have observed that when $b_v$ is set to 10 to 100 (i.e., default to 10 in experiments), it can achieve good verification results and find the important data

regions with low overhead. All reported results in Table 1 are from the recommended hyperparameters of STAFF. In addition, we discuss and analyze the impact of hyperparameters (e.g., $b_v$ and $T$) on the effect of STAFF in §A.4.

**Table 4:** The performance of the models on downstream tasks without and with fine-tuning.

| Model | BioInstruct | | DialogSum | | WMT-19 | |
|---|---|---|---|---|---|---|
| | BLEU-4 | Rouge-L | BLEU-4 | Rouge-L | BLEU-4 | Rouge-L |
| **Gemma-7b** | 8.3(47.4) | 6.9(41.8) | 6.2(52.8) | 7.7(49.8) | 0.2(68.8) | 0.2(62.2) |
| **Gemma-2b** | 5.1(44.8) | 4.8(39.4) | 3.7(50.8) | 5.8(47.7) | 0.3(61.2) | 0.3(53.4) |
| **Llama-2-13b** | 5.5(46.8) | 4.8(41.7) | 1.6(52.5) | 2.2(49.6) | 0.3(67.9) | 0.2(62.1) |
| **Llama-2-7b** | 3.0(46.3) | 2.8(41.0) | 2.8(51.8) | 3.9(49.3) | 2.3(65.7) | 1.8(59.0) |
| **Mistral-Nemo** | 14.9(48.4) | 13.7(43.2) | 20.9(53.3) | 21.5(50.3) | 6.9(71.0) | 7.9(65.8) |
| **Mistral-7B** | 17.2(48.0) | 17.0(42.4) | 21.0(52.6) | 22.2(49.9) | 8.0(69.6) | 8.0(63.6) |

**Model and Training.** In our experiments, we evaluate STAFF on Gemma-7b, Llama-2-13b, and Mistral-Nemo (12b) and use small models Gemma-2b, Llama-2-7b, and Mistral-7B from their family to calculate the speculative score. The model performance without and with fine-tuning on the given task is shown in Table 4. We use gray to mark the performance of the fine-tuned models.

For fine-tuning pre-trained models on three datasets of downstream tasks, we perform a grid search over learning rate $\{1e^{-5}, 2e^{-5}, 1e^{-4}, 2e^{-4}\}$ and the batch size $\{2, 4, 8\}$, using the complete dataset. The results in the batch size of 8 and learning rates are shown in Table 5. Models trained on pruned datasets or full datasets use the same hyperparameters. All fine-tunings are conducted with the parameter-efficient fine-tuning method 'LoRA' (Hu et al., 2022). We use the fine-tuning framework from Zheng et al. (2024) to conduct fine-tuning and employ a learning rate scheduler with linear warm-up and cosine decay. We have initially conducted large-scale experiments to determine the number of training epochs for the LLMs. We fine-tune all models on the full dataset for up to 50 epochs. However, we have observed that after certain epochs of fine-tuning, increasing the number of epochs no longer improves model performance. Consequently, we opt for a fixed number of epochs (e.g., 4 epochs) in all experiments. All the fine-tunings in the experiments are repeated three times with different random seeds and we report the averaged results in §4.

**Table 5:** Learning rate for each model on different datasets.

| Model | Dataset | | |
|---|---|---|---|
| | BioInstruct | DialogSum | WMT-19 |
| Gemma-7b | 1e-4 | 1e-4 | 2e-4 |
| Llama-2-13b | 2e-5 | 2e-4 | 2e-4 |
| Mistral-Nemo | 1e-5 | 1e-4 | 2e-4 |

**Theoretical Analysis of Effort Score.** We use the effort score proposed and widely used in prior work (Paul et al., 2021; Fayyaz et al., 2022; Lin et al., 2024), to calculate the speculative score, which measures the effort of models to learn and fit a specific data sample. For the data sample $d$ consisting of $(x, y)$, where $x$ is the input (i.e., a sequence of tokens) and $y$ is the corresponding label. The model gradient at time $t$ of this sample can be represented as follows.

$$g_t(x, y) = \nabla_\phi L(\theta^t(x, y))). \tag{5}$$

According to Lemma 2.2 and corresponding proof of Paul et al. (2021), for all $d^* = (x^*, y^*)$ there exists a constant $c$ that satisfies.

$$\|\Delta_t((x^*, y^*), B) - \Delta_t((x^*, y^*), B')\|_2 \leq c\|g_t(x_j, y_j))\|_2, \tag{6}$$

where $B$ represents the minibatch of in model training and $B' = B \backslash \{d_j = (x_j, y_j)\}$. $\Delta_t((x^*, y^*), B)$ is the time derivative of the loss for the example $x^*, y^*$. Equation 6 indicates that the contribution of sample $d_j$ to the loss is constrained by $\|g_t(x_j, y_j))\|_2$. When the norm of the gradient $\|g_t(x_j, y_j))\|_2$ is smaller, the model has to make less effort to learn and fit the sample, and the removal of the data has less impact on the model training. Therefore, such a gradient norm can be used to evaluate the

importance and contribution of data samples to model training. In §3.1, the fine-tuned model $\theta_s$ is no longer updated with time $t$, and the speculation score $S_d^s$ (i.e., effort score) can be expressed as:

$$S_d^s = \text{Effort}(x, y) = \|\nabla_\phi L(\theta_s(x, y)))\|_2.\tag{7}$$

Effort Score has been widely used in existing work (Fayyaz et al., 2022; Lin et al., 2024). Following the open-sourced code of these works, we have implemented Equation 7 in our repository.

### A.4 ADDITIONAL RESULTS.

**Results of Other Metrics.** Following prior work (Zheng et al., 2024; Ko et al., 2023; Zhang et al., 2024a; Lin et al., 2024), we use the metrics Rouge-L and BLEU-4 to evaluate the performance of coreset selection methods on downstream tasks. As a supplement to §4, we provide the experiment results and the standard deviation (marked by gray) of each coreset selection method on all metrics.

Table 6, Table 8 and Table 10 illustrate that the observations in §4.2 still hold for the BLEU metric. STAFF can achieve better performance than the SOTA methods across different pruning rates, which shows the effectiveness of STAFF in coreset selection across different pruning rates.

**Table 6:** Performance (BLEU-4) of STAFF and baselines on the BioInstruct dataset. (Higher is better. Bold indicates the best result, italic and underlined indicate the second-best result.)

| Model | Method | Pruning Rate | | | | | |
|---|---|---|---|---|---|---|---|
| | | 0% | 20% | 50% | 70% | 80% | 90% |
| **Gemma-7b** | Random | 47.4(0.4) | 46.6(0.3) | 46.4(0.4) | 45.7(0.3) | 45.2(0.5) | 45.1(0.2) |
| | GraNd | - | 46.9(0.6) | 46.4(0.4) | 45.4(0.5) | 43.8(0.4) | 42.8(0.5) |
| | EL2N | - | 47.2(0.4) | *47.7(0.5)* | *46.3(0.5)* | *45.8(0.2)* | 43.9(0.4) |
| | CCS | - | *47.4(0.4)* | 47.2(0.3) | 46.2(0.1) | 45.8(0.4) | *45.2(0.4)* |
| | D2 Pruning | - | 47.3(0.1) | 47.0(0.2) | 46.0(0.3) | 45.4(0.4) | 44.5(0.5) |
| | STAFF | - | **48.4(0.3)** | **47.8(0.3)** | **47.4(0.4)** | **47.3(0.3)** | **46.2(0.1)** |
| **Llama-2-13b** | Random | 46.8(0.3) | 45.9(0.3) | 45.3(0.4) | 44.7(0.5) | 44.1(0.5) | 43.5(0.6) |
| | GraNd | - | 46.4(0.2) | 43.9(0.2) | 41.9(0.2) | 40.3(0.3) | 36.8(0.3) |
| | EL2N | - | 46.2(0.6) | 45.8(0.1) | *45.9(0.4)* | *45.0(0.4)* | 44.2(0.3) |
| | CCS | - | *46.8(0.3)* | *45.9(0.2)* | 45.3(0.3) | 44.5(0.3) | 43.5(0.3) |
| | D2 Pruning | - | 45.9(0.2) | 45.5(0.5) | 44.9(0.3) | 45.0(0.2) | *44.5(0.3)* |
| | STAFF | - | **47.6(0.5)** | **47.1(0.5)** | **46.5(0.3)** | **46.2(0.3)** | **45.5(0.4)** |
| **Mistral-Nemo** | Random | 48.4(0.7) | 48.2(0.3) | 46.9(0.4) | 46.4(0.1) | 46.3(0.4) | 45.9(0.4) |
| | GraNd | - | 47.7(0.2) | 46.0(0.3) | 44.7(0.6) | 43.1(0.3) | 41.0(0.4) |
| | EL2N | - | **48.9(0.2)** | 48.3(0.5) | *47.1(0.3)* | 46.7(0.3) | 45.4(0.3) |
| | CCS | - | 48.7(0.2) | 47.8(0.4) | 46.9(0.4) | **47.2(0.3)** | 45.1(0.4) |
| | D2 Pruning | - | 48.7(0.1) | *48.3(0.3)* | 46.6(0.1) | 46.6(0.3) | 44.7(0.2) |
| | STAFF | - | *48.7(0.4)* | **48.3(0.4)** | **48.0(0.2)** | *47.1(0.5)* | **46.3(0.2)** |

**Table 7:** Performance (Rouge-L) of STAFF and baselines on the BioInstruct dataset. (Higher is better. Bold indicates the best result, italic and underlined indicate the second-best result.)

| Model | Method | Pruning Rate | | | | | |
|---|---|---|---|---|---|---|---|
| | | 0% | 20% | 50% | 70% | 80% | 90% |
| **Gemma-7b** | Random | 41.8(0.1) | 41.1(0.3) | 40.6(0.3) | 39.5(0.2) | 38.8(0.4) | 38.5(0.2) |
| | GraNd | - | 40.3(0.5) | 39.7(0.1) | 39.1(0.2) | 37.9(0.3) | 37.8(0.1) |
| | EL2N | - | 41.3(0.5) | 41.3(0.5) | 40.4(0.4) | 39.8(0.3) | 38.8(0.3) |
| | CCS | - | *41.7(0.3)* | *41.4(0.3)* | 40.3(0.3) | *40.6(0.4)* | *39.3(0.3)* |
| | D2 Pruning | - | 41.4(0.2) | 41.2(0.3) | *40.5(0.4)* | 40.2(0.1) | 39.1(0.4) |
| | STAFF | - | **42.0(0.5)** | **41.5(0.1)** | **41.3(0.4)** | **40.8(0.2)** | *39.4(0.5)* |
| **Llama-2-13b** | Random | 41.7(0.3) | 40.9(0.5) | 40.3(0.2) | 39.5(0.4) | 38.6(0.5) | 37.9(0.3) |
| | GraNd | - | 40.5(0.3) | 39.1(0.5) | 37.8(0.5) | 36.9(0.2) | 35.2(0.3) |
| | EL2N | - | 40.7(0.1) | 39.8(0.1) | 39.6(0.2) | 38.9(0.2) | *38.5(0.1)* |
| | CCS | - | *41.7(0.3)* | *41.0(0.3)* | *40.5(0.2)* | *39.9(0.1)* | 38.5(0.3) |
| | D2 Pruning | - | 41.4(0.2) | 40.7(0.3) | 40.2(0.1) | 39.7(0.2) | 38.4(0.3) |
| | STAFF | - | **42.2(0.4)** | **41.6(0.3)** | **40.7(0.1)** | **40.2(0.6)** | **39.8(0.2)** |
| **Mistral-Nemo** | Random | 43.2(0.2) | 42.8(0.3) | 41.3(0.2) | 40.6(0.3) | 40.5(0.6) | 40.2(0.5) |
| | GraNd | - | 41.5(0.1) | 40.2(0.3) | 39.5(0.3) | 38.6(0.3) | 37.8(0.2) |
| | EL2N | - | **43.2(0.6)** | 42.3(0.1) | 41.4(0.7) | 41.2(0.2) | 40.2(0.3) |
| | CCS | - | 43.0(0.2) | *42.5(0.4)* | *42.2(0.3)* | **41.9(0.4)** | *40.6(0.6)* |
| | D2 Pruning | - | 42.9(0.4) | **42.7(0.4)** | 41.2(0.1) | 41.0(0.4) | 40.0(0.5) |
| | STAFF | - | *43.1(0.3)* | 42.3(0.4) | **42.2(0.3)** | *41.2(0.4)* | **41.2(0.7)** |

**Table 8:** Performance (BLEU-4) of STAFF and baselines on the DialogSum dataset. (Higher is better. Bold indicates the best result, italic and underlined indicate the second-best result.)

| Model | Method | Pruning Rate | | | | | |
|---|---|---|---|---|---|---|---|
| | | 0% | 20% | 50% | 70% | 80% | 90% |
| **Gemma-7b** | Random | 52.8(0.2) | 51.2(0.4) | 50.0(0.6) | 49.8(0.2) | 49.6(0.5) | 48.5(0.3) |
| | GraNd | - | 51.9(0.1) | 50.9(0.3) | 49.6(0.4) | 48.7(0.3) | 47.1(0.2) |
| | EL2N | - | 52.1(0.4) | 51.5(0.4) | 51.1(0.1) | 50.4(0.2) | 49.4(0.3) |
| | CCS | - | *52.3(0.5)* | 51.1(0.5) | *51.7(0.5)* | *50.9(0.4)* | *50.3(0.4)* |
| | D2 Pruning | - | 52.2(0.4) | *51.6(0.4)* | 50.9(0.7) | 50.3(0.4) | 49.2(0.4) |
| | STAFF | - | **53.1(0.2)** | **52.0(0.1)** | **52.1(0.3)** | **51.4(0.2)** | **51.1(0.4)** |
| **Llama-2-13b** | Random | 52.5(0.1) | 51.8(0.2) | 50.7(0.4) | 49.3(0.4) | 48.5(0.6) | 48.1(0.4) |
| | GraNd | - | 52.2(0.6) | 49.8(0.4) | 47.8(0.4) | 46.4(0.3) | 43.0(0.2) |
| | EL2N | - | *52.7(0.3)* | **51.9(0.3)** | 51.3(0.3) | *50.5(0.4)* | 49.2(0.4) |
| | CCS | - | 51.9(0.2) | *51.9(0.1)* | 50.9(0.6) | 50.0(0.5) | *49.6(0.2)* |
| | D2 Pruning | - | 51.7(0.4) | 51.0(0.4) | 50.7(0.4) | 50.0(0.4) | 49.1(0.3) |
| | STAFF | - | **52.8(0.4)** | 51.9(0.3) | **51.7(0.4)** | **51.0(0.4)** | **50.1(0.2)** |
| **Mistral-Nemo** | Random | 53.3(0.1) | 52.5(0.3) | 51.4(0.4) | 51.2(0.5) | 50.1(0.3) | 49.6(0.1) |
| | GraNd | - | 52.8(0.1) | 51.6(0.5) | 49.7(0.1) | 48.8(0.2) | 46.2(0.2) |
| | EL2N | - | *53.4(0.3)* | 51.4(0.5) | 50.1(0.3) | 49.7(0.3) | 48.6(0.6) |
| | CCS | - | **53.5(0.4)** | *52.0(0.8)* | **52.1(0.1)** | *51.3(0.8)* | *50.3(0.4)* |
| | D2 Pruning | - | 53.0(0.5) | 51.8(0.2) | 50.4(0.7) | 49.7(0.4) | 48.6(0.3) |
| | STAFF | - | 53.2(0.1) | **52.7(0.4)** | *51.9(0.4)* | **52.1(0.6)** | **51.5(0.2)** |

**Table 9:** Performance (Rouge-L) of STAFF and baselines on the DialogSum dataset. (Higher is better. Bold indicates the best result, italic and underlined indicate the second-best result.)

| Model | Method | Pruning Rate | | | | | |
|---|---|---|---|---|---|---|---|
| | | 0% | 20% | 50% | 70% | 80% | 90% |
| Gemma-7b | Random | 49.8(0.3) | 48.0(0.5) | 47.0(0.4) | 47.0(0.2) | 46.8(0.4) | 45.5(0.6) |
| | GraNd | - | 48.7(0.4) | 47.8(0.4) | 46.8(0.4) | 46.6(0.6) | 45.5(0.4) |
| | EL2N | - | 49.5(0.3) | 48.6(0.3) | 47.7(0.2) | 47.0(0.5) | 46.2(0.3) |
| | CCS | - | *49.6(0.4)* | 48.1(0.2) | *48.3(0.4)* | *48.2(0.7)* | *47.4(0.3)* |
| | D2 Pruning | - | 49.4(0.4) | *48.7(0.4)* | 47.6(0.4) | 46.9(0.4) | 45.9(0.5) |
| | STAFF | - | **50.0(0.6)** | **48.9(0.6)** | **48.9(0.2)** | **48.5(0.3)** | **48.4(0.6)** |
| Llama-2-13b | Random | 49.6(0.5) | 49.1(0.4) | 48.1(0.3) | 46.9(0.2) | 46.2(0.6) | 45.5(0.1) |
| | GraNd | - | *49.3(0.3)* | 48.2(0.2) | 46.5(0.4) | 45.4(0.2) | 44.7(0.3) |
| | EL2N | - | 48.9(0.4) | 48.5(0.4) | 47.7(0.5) | 47.3(0.3) | 45.9(0.3) |
| | CCS | - | 48.9(0.3) | *49.1(0.6)* | *48.7(0.4)* | *48.0(0.4)* | *47.5(0.2)* |
| | D2 Pruning | - | 48.8(0.2) | 48.4(0.8) | 48.0(0.2) | 47.0(0.6) | 46.1(0.1) |
| | STAFF | - | **50.0(0.4)** | **49.1(0.6)** | **49.0(0.3)** | **48.4(0.3)** | **48.2(0.3)** |
| Mistral-Nemo | Random | 50.3(0.3) | 49.7(0.5) | 48.3(0.1) | 48.2(0.1) | 47.1(0.3) | 47.1(0.4) |
| | GraNd | - | 49.6(0.7) | *49.2(0.4)* | 47.9(0.6) | 47.5(0.4) | 45.9(0.3) |
| | EL2N | - | 50.3(0.3) | 48.4(0.1) | 47.8(0.3) | 46.9(0.5) | 46.4(0.2) |
| | CCS | - | **50.7(0.4)** | 49.0(0.2) | **49.5(0.5)** | *49.0(0.2)* | *48.7(0.3)* |
| | D2 Pruning | - | 50.2(0.2) | 48.8(0.7) | 47.6(0.4) | 47.1(0.3) | 46.3(0.3) |
| | STAFF | - | *50.4(0.3)* | **50.2(0.5)** | *49.2(0.6)* | **49.2(0.4)** | **48.9(0.2)** |

**Table 10:** Performance (BLEU-4) of STAFF and baselines on the WMT-19 dataset. (Higher is better. Bold indicates the best result, italic and underlined indicate the second-best result.)

| Model | Method | Pruning Rate | | | | | |
|---|---|---|---|---|---|---|---|
| | | 0% | 20% | 50% | 70% | 80% | 90% |
| Gemma-7b | Random | 68.8(0.4) | 67.5(0.6) | 65.5(0.3) | 63.8(0.2) | 62.8(0.4) | 61.2(0.4) |
| | GraNd | - | 66.8(0.3) | 60.3(0.1) | 56.7(0.2) | 53.7(0.1) | 49.3(0.3) |
| | EL2N | - | 67.7(0.5) | 62.5(0.5) | 56.0(0.1) | 50.8(0.4) | 50.1(0.2) |
| | CCS | - | 68.0(0.3) | *66.5(0.6)* | *64.6(0.3)* | *62.9(0.2)* | *61.4(0.3)* |
| | D2 Pruning | - | *68.1(0.4)* | 64.6(0.5) | 59.5(0.1) | 57.0(0.6) | 58.5(0.4) |
| | STAFF | - | **68.7(0.6)** | **66.8(0.5)** | **65.4(0.5)** | **64.2(0.1)** | **63.3(0.5)** |
| Llama-2-13b | Random | 67.9(0.2) | 66.4(0.5) | 64.5(0.1) | 62.3(0.6) | 61.1(0.3) | 58.9(0.6) |
| | GraNd | - | 64.6(0.9) | 58.5(0.6) | 51.8(0.5) | 48.1(0.3) | 42.5(0.5) |
| | EL2N | - | 65.9(0.2) | 61.2(0.6) | 56.3(0.5) | 54.2(0.5) | 49.6(0.2) |
| | CCS | - | 67.3(0.6) | *65.7(0.2)* | **63.7(0.9)** | *62.7(0.2)* | *60.5(0.4)* |
| | D2 Pruning | - | *67.3(0.5)* | 65.0(0.3) | 62.2(0.6) | 59.1(0.3) | 56.2(0.3) |
| | STAFF | - | **67.8(0.2)** | **65.7(0.2)** | *63.5(0.3)* | **62.8(0.6)** | **61.0(0.5)** |
| Mistral-Nemo | Random | 71.0(0.5) | 70.1(0.4) | 68.2(0.6) | 66.9(0.2) | 65.9(0.7) | 64.6(0.9) |
| | GraNd | - | 69.6(0.5) | 64.9(0.3) | 59.2(0.4) | 57.7(0.3) | 52.3(0.4) |
| | EL2N | - | 70.3(0.4) | 68.0(0.2) | 65.5(0.3) | 64.0(0.4) | 60.5(0.7) |
| | CCS | - | *70.4(0.5)* | *69.3(0.7)* | *68.0(0.2)* | *67.4(0.4)* | *65.8(0.2)* |
| | D2 Pruning | - | 70.4(0.8) | 68.3(0.4) | 66.2(0.4) | 64.3(0.7) | 61.7(0.4) |
| | STAFF | - | **70.8(0.6)** | **69.8(0.4)** | **68.8(0.6)** | **67.9(0.5)** | **66.1(0.3)** |

**Table 11:** Performance (Rouge-L) of STAFF and baselines on the WMT-19 dataset. (Higher is better. Bold indicates the best result, italic and underlined indicate the second-best result.)

| Model | Method | Pruning Rate | | | | | |
|---|---|---|---|---|---|---|---|
| | | 0% | 20% | 50% | 70% | 80% | 90% |
| Gemma-7b | Random | 62.2(0.3) | 61.0(0.4) | 58.8(0.9) | 57.0(0.4) | 55.7(0.4) | 53.7(0.4) |
| | GraNd | - | 60.2(0.3) | 52.4(0.4) | 48.7(0.6) | 45.1(0.4) | 40.3(0.3) |
| | EL2N | - | 61.1(0.3) | 55.6(0.4) | 51.2(0.3) | 46.6(0.6) | 44.9(0.6) |
| | CCS | - | *61.6(0.1)* | *59.7(0.8)* | *57.4(0.4)* | *55.8(0.3)* | *54.1(0.4)* |
| | D2 Pruning | - | 61.5(0.3) | 57.5(0.7) | 53.7(0.6) | 51.6(0.1) | 51.4(0.4) |
| | STAFF | - | **62.4(0.5)** | **59.9(0.2)** | **58.2(0.5)** | **56.9(0.1)** | **55.9(0.3)** |
| Llama-2-13b | Random | 62.1(0.7) | 60.6(0.4) | 58.4(0.6) | 55.8(0.4) | 54.5(0.3) | 52.1(0.5) |
| | GraNd | - | 58.7(0.2) | 51.8(0.1) | 44.8(0.3) | 40.2(0.4) | 35.5(0.3) |
| | EL2N | - | 60.0(0.5) | 55.1(0.2) | 50.6(0.5) | 48.3(0.3) | 42.8(0.3) |
| | CCS | - | 61.2(0.3) | *59.3(0.5)* | *57.2(0.4)* | *56.0(0.4)* | *53.6(0.5)* |
| | D2 Pruning | - | *61.4(0.5)* | 58.7(0.7) | 55.8(0.2) | 52.4(0.5) | 49.4(0.4) |
| | STAFF | - | **62.0(0.6)** | **59.5(0.5)** | **56.9(0.5)** | **56.2(0.2)** | **54.8(0.4)** |
| Mistral-Nemo | Random | 65.8(0.1) | 64.8(0.2) | 62.5(0.4) | 61.1(0.4) | 59.8(0.9) | 58.4(0.3) |
| | GraNd | - | 64.4(0.1) | 59.4(0.3) | 53.2(0.4) | 50.5(0.1) | 44.9(0.6) |
| | EL2N | - | 64.7(0.2) | 61.5(0.4) | 58.3(0.4) | 56.0(0.2) | 51.9(0.4) |
| | CCS | - | *65.1(0.5)* | *63.5(1.0)* | *61.8(0.1)* | *61.3(0.6)* | *59.2(0.5)* |
| | D2 Pruning | - | 65.0(0.6) | 61.9(0.5) | 59.1(0.4) | 56.8(0.5) | 53.6(0.2) |
| | STAFF | - | **65.4(0.6)** | **64.1(0.2)** | **63.1(0.5)** | **62.0(0.4)** | **60.3(0.5)** |

**Fine-tuning Budget in Coreset Selection.** $T$ represents the fine-tuning budget in coreset selection. Existing methods typically require fine-tuning the model for several epochs (i.e., $T$) to evaluate data scores or divide data regions. Based on the experiment results in prior work (Paul et al., 2021), we set $T = 3$ in our experiments. To understand the impact of $T$ on the performance of selection results, we conduct experiments on the Gemma-7b model and WMT-19 dataset, and the results (Rouge-L) are shown in Table 12. We observe that increasing $T$ generally improves the coreset performance for the selection methods. This is because a larger $T$ helps the selection method accurately evaluate and identify important samples and data regions. For example, when the pruning rate is 90%, the selection result of the GraNd method improves from 41.0 to 44.9 when T increases from 1 to 5. STAFF also improves from 55.9 to 56.6 during this process. Moreover, STAFF achieves the best results in all $T$ settings, further demonstrating the effectiveness of STAFF in coreset selection.

**Table 12:** Impact of fine-tuning budget $T$ on selection results (Rouge-L). (Higher is better. Bold indicates the best result, italic and underlined indicate the second-best result)

| Method | T=1 | | | | | T=3 | | | | | T=5 | | | | |
|---|---|---|---|---|---|---|---|---|---|---|---|---|---|---|---|
| | 20% | 50% | 70% | 80% | 90% | 20% | 50% | 70% | 80% | 90% | 20% | 50% | 70% | 80% | 90% |
| GraNd | 59.3 | 54.0 | 48.4 | 45.1 | 41.0 | 60.2 | 52.4 | 48.7 | 45.1 | 40.3 | 61.1 | 55.8 | 50.7 | 46.6 | 44.9 |
| EL2N | 61.0 | 57.8 | 54.0 | 51.1 | 46.7 | 61.1 | 55.6 | 51.2 | 46.6 | 44.9 | 61.4 | 58.0 | 53.6 | 51.3 | 47.9 |
| CCS | 61.2 | _59.5_ | _57.2_ | _55.7_ | _52.8_ | _61.6_ | _59.7_ | _57.4_ | _55.8_ | _54.1_ | 61.1 | _59.7_ | _58.2_ | _56.2_ | _54.1_ |
| D2 Pruning | _61.4_ | 58.7 | 56.2 | 54.2 | 51.0 | 61.5 | 57.5 | 53.7 | 51.6 | 51.4 | _61.8_ | 58.8 | 55.2 | 53.1 | 51.1 |
| STAFF | **61.9** | **60.3** | **58.7** | **56.9** | **55.9** | **62.4** | **59.9** | **58.2** | **56.9** | **55.9** | **62.2** | **60.7** | **58.6** | **57.6** | **56.6** |

**Speculative Scoring Function.** We employ the effort score from Paul et al. (2021) as the scoring function for the small model to assess the changes the model makes when learning each input sample. A larger effort score indicates a greater discrepancy between the sample and the knowledge distribution encoded in the model. Other scoring functions also have the potential to be used as speculative scoring functions. Table 13 presents a comparison of fine-tuning results (Rouge-L) using the importance score (Paul et al., 2021) and influence score (Koh & Liang, 2017) on the WMT-19 dataset and the Gemma-7b model at different pruning rate. All scores can be easily obtained on a smaller model, and the configurations in the experiment are consistent with §4.1. We can observe that these scoring functions bring certain performance degradation. However, they are still better than several baseline methods and almost do not suffer from catastrophic performance degradation at high pruning rates, which illustrates the effectiveness of LLM verification and selection in STAFF.

**Table 13:** Impact of selection scoring function $T$ on selection results (Rouge-L).

| Scoring Function | WMT-19 | | | | |
|---|---|---|---|---|---|
| | 20% | 50% | 70% | 80% | 90% |
| Effort Score | 62.4 | 59.9 | 58.2 | 56.9 | 55.9 |
| Importance Score | 59.4 | 58.2 | 56.5 | 55.0 | 53.0 |
| Influence Score | 60.5 | 58.8 | 57.8 | 56.8 | 55.2 |

**Verification Budget.** $b_v$ is the verification budget. A larger $b_v$ brings more accurate verification and evaluation on the bin $B_i$ and greater overhead in verification. We conduct experiments on the WMT-19 dataset and the Gemma-7b model to show the impact of the validation budget $b_v$ on the results. We can observe that for medium to low pruning rates, increasing the budget does not significantly improve or degrade the results. This is because most data is still retained in the coreset, and the validation budget has little impact on the coreset selection. For high pruning rates, we observe that as the validation budget increases, the performance of the selected coreset generally improves. In this case, a higher validation budget allows for a more accurate evaluation of the importance of each data region to the target model, enabling adjustments to the selection budget to avoid missing important data regions. We suggest selecting $b_v \in [10, 100]$ based on the available budget.

**Table 14:** Impact of verification budget $b_v$ on selection results (Rouge-L).

| $b_v$ | WMT-19 | | | | |
|---|---|---|---|---|---|
| | 20% | 50% | 70% | 80% | 90% |
| **1** | 61.4 | 60.3 | 58.1 | 56.6 | 54.7 |
| **10** | 62.4 | 59.9 | 58.2 | 56.9 | 55.9 |
| **100** | 61.5 | 59.3 | 58.4 | 57.4 | 56.0 |
| $|B_i|$ | 61.7 | 59.2 | 58.4 | 57.3 | 56.2 |

**Ablation Study of Small Models.** To further study the impact of small models of different sizes on the coreset selection results, we conduct experiments on the Llama-2-13b model and the DialogSum dataset. Since the Llama-2-7b used in our experiment is already the smallest official model in its family, we use two pruned versions of Llama-2-7b as the speculative models in this experiment, with 50% and 70% parameters pruned by SparseGPT (Frantar & Alistarh, 2023). The coreset selection results are shown in Table 15. We can observe that both pruned models face performance degradation compared with the original Llama-2-7b model. The performance degradation of the model with a 70% pruning rate is more significant than that of the model with a 50% pruning rate. In addition, the models with 50% and 70% pruning rates do not bring significant improvements in efficiency and only reduce the coreset selection time by 2.4% and 3.8%, respectively. Although smaller models (models with more parameters pruned) have the potential to significantly reduce fine-tuning and selection costs, they cannot effectively obtain a data score distribution similar to that of the target model, leading to a degradation in selection performance. There is a trade-off between data selection effectiveness and efficiency. We currently recommend that users directly use officially released pre-trained small models from the same family to calculate speculative scores.

Note that the effectiveness of STAFF depends on the similarity of knowledge and functions of the small model and the target model. Two similar models have a greater chance of obtaining similar data score distributions, thereby achieving better coreset selection results. However, existing work lacks a comprehensive and effective metric to evaluate the similarity of functions and performance between models. Wang et al. (2024b) have proposed a series of metrics to evaluate the representational (functional) similarity between LLMs, but they observe that the evaluation results of different metrics varied greatly and are difficult to interpret. Building a framework to evaluate the similarity between large and small models would help STAFF select effective and efficient small models in the future.

**Table 15:** Impact of different small models on the selection results.

| Method | ROUGE-L | | | | | BLEU-4 | | | | |
|---|---|---|---|---|---|---|---|---|---|---|
| | 20% | 50% | 70% | 80% | 90% | 20% | 50% | 70% | 80% | 90% |
| STAFF +Llama-2-7b | **50.0** | 49.1 | **49.0** | 48.4 | **48.2** | 52.8 | 51.9 | 51.7 | 51.0 | 50.1 |
| STAFF +Llama-2-7b-Pruned 50% | 49.9 | 49.1 | 49.0 | **48.5** | 48.1 | 52.6 | 51.6 | 51.6 | 50.9 | **50.2** |
| STAFF +Llama-2-7b-Pruned 70% | 49.6 | 49.1 | 48.9 | 48.2 | 47.7 | 52.4 | 51.3 | 51.3 | 50.9 | 50.1 |
| Random | 49.1 | 48.1 | 46.9 | 46.2 | 45.5 | 51.8 | 50.7 | 49.3 | 48.5 | 48.1 |

**Effectiveness of STAFF on Larger Models.** Due to the limitation of computation resources, we use small models to select coresets for the large models with less than 13 billion parameters in our experiments. Existing speculative decoding work (Miao et al., 2024) has demonstrated that the small model (e.g., LLaMA-68M) can guide and accelerate the inference of a much larger model in the same family (LLaMA-65B). STAFF, which is also built on the concept of speculative execution, can theoretically use a small model to guide data selection for a larger model (e.g., 32B).

To demonstrate the effectiveness of STAFF in guiding the coreset selection of larger target models, we conduct experiments on the DialogSum dataset with the Qwen2.5-32B model (Team, 2024). In experiments, we separately use Qwen2.5-3B, 7B, and 14B as the speculative model to study the impact of different speculative models. Similar to the results in Table 4, these Qwen2.5 models have achieved significant performance improvements after fine-tuning. For example, the ROUGE-L of Qwen2.5-7B has been increased from 23.0 to 52.5 after fine-tuning. The experimental results at different sampling rates and metrics are shown in Table 16. We can observe that for the target model with over 30B parameters, STAFF can still use the small model to effectively guide coreset selection, which is consistent with the above theoretical analysis. Even using the 3B version to evaluate

speculative scores, STAFF can still effectively select data samples and achieve better fine-tuning results than baselines at different pruning rates.

In addition, due to the much smaller size of the speculative model and the much lower fine-tuning overhead than the target model, the 3B model only takes 3.0 hours for data selection (p=90%), reducing the selection cost by 88.5% to 89.7% compared to baseline methods, which further demonstrates the efficiency and effectiveness of STAFF in coreset selection. In comparison, although the 7B and 14B models can achieve overall better selection results, the time overhead of fine-tuning these models and selecting coresets has also increased, reaching 3.1 and 5.3 hours, respectively. For speculative models of the same family, a larger model has the potential to have a more similar knowledge distribution to the target model and achieve better selection results, but it will also bring greater selection overhead. For example, the selection overhead of the 14B version is 1.77 times that of the 3B version. Considering the limited performance improvement brought by such a significant increase in selection overhead, we recommend users choose the smallest possible officially released model in the model family (e.g., those with a size of less than 7B) as the speculative model.

**Table 16:** Performance of STAFF and baselines on the Qwen-2.5-32B model. (Higher is better. Bold indicates the best result, italic and underlined indicate the second-best result)

| Model | Method | ROUGE-L | | | | | | BLEU-4 | | | | | |
|---|---|---|---|---|---|---|---|---|---|---|---|---|---|
| | | 0% | 20% | 50% | 70% | 80% | 90% | 0% | 20% | 50% | 70% | 80% | 90% |
| Qwen2.5-32B | Random | 50.2 | 49.5 | 48.6 | 48.1 | 47.5 | 47.1 | 53.2 | 52.4 | 51.4 | 51.1 | 50.4 | 49.2 |
| | GraNd | - | 50.5 | 48.6 | 47.5 | 46.9 | 45.6 | - | 52.8 | 50.9 | 49.1 | 47.2 | 45.5 |
| | EL2N | - | 50.5 | 48.9 | 48.2 | 47.6 | 46.3 | - | 52.8 | 51.8 | 50.5 | 50.1 | 47.9 |
| | CCS | - | 50.2 | 49.3 | 49.1 | 49.0 | 48.6 | - | 52.8 | 52.1 | 52.1 | 51.6 | 51.3 |
| | D2 Pruning | - | 49.7 | 49.7 | 48.9 | 47.2 | 46.8 | - | 52.0 | 52.4 | 50.9 | 49.4 | 48.2 |
| | STAFF-3B | - | 50.5 | 50.0 | **49.7** | 49.0 | 48.9 | - | 53.3 | 52.7 | **52.6** | 51.5 | 51.2 |
| | STAFF-7B | - | _50.5_ | **50.2** | 49.6 | _49.2_ | _49.1_ | - | _53.4_ | **53.0** | 52.3 | **51.9** | _51.6_ |
| | STAFF-14B | - | **51.1** | _50.1_ | _49.6_ | **49.3** | **49.1** | - | **53.7** | _52.9_ | _52.4_ | _51.5_ | **51.6** |

**Effectiveness of STAFF on Other Tasks.** To demonstrate the effectiveness of STAFF in guiding the coreset selection on diverse tasks, we conduct additional experiments with the Gemma-7b model on the subset of the Ltg-En dataset (Group, 2024). This dataset is collected from Wikipedia by the Language Technology Group at the University of Oslo. It contains information from Wikipedia and corresponding more natural, human-friendly versions of the paraphrases. We use a subset of this dataset with 19,999 items in the experiment. Similar to the results in Table 4, the Gemma models have achieved significant performance improvements after fine-tuning, e.g., the ROUGE-L of Gemma-2b is increased from 13.3 to 82.0. The experimental results at different sampling rates and metrics are shown in Table 17. We can observe that in the paraphrasing task, the coreset selection effect of STAFF still outperforms the baselines at different pruning rates, and improves the baseline results by up to 65.72%. Note that, when $p = 20\%$, the coreset selected by STAFF achieves better fine-tuning effects than the complete dataset on both ROUGE-L and BLEU-4 metrics, which further illustrates the effectiveness of STAFF in selecting coresets and improving data efficiency for LLMs on various tasks. In addition, the selection overhead of STAFF is only 2.4 hours (p=90%), which is 58.7% to 72.3% shorter than that of the baseline methods. We will update the results as soon as the experiment is finished.

**Table 17:** Performance of STAFF and baselines on the Ltg-En dataset on the paraphrasing task. (Higher is better. Bold indicates the best result, italic and underlined indicate the second-best result)

| Model | Method | ROUGE-L | | | | | | BLEU-4 | | | | | |
|---|---|---|---|---|---|---|---|---|---|---|---|---|---|
| | | 0% | 20% | 50% | 70% | 80% | 90% | 0% | 20% | 50% | 70% | 80% | 90% |
| Gemma-7b | Random | 82.5 | 81.9 | 81.3 | 80.7 | 80.9 | 80.6 | 87.4 | 86.9 | 86.3 | 85.7 | 85.8 | 85.6 |
| | GraNd | - | 81.9 | 77.0 | 58.6 | 49.7 | 49.3 | - | 87.1 | 83.4 | 69.7 | 63.1 | 62.9 |
| | EL2N | - | 80.9 | 69.2 | 63.7 | 64.2 | 64.6 | - | 86.3 | 77.7 | 73.5 | 73.8 | 74.2 |
| | CCS | - | _82.2_ | _81.6_ | _80.7_ | _81.1_ | _80.5_ | - | _87.2_ | _86.8_ | _86.1_ | _86.3_ | _85.9_ |
| | D2 Pruning | - | 81.6 | 70.7 | 64.7 | 63.2 | 64.5 | - | 86.7 | 78.8 | 74.2 | 73.2 | 73.9 |
| | STAFF | - | **83.0** | **81.9** | **81.9** | **81.8** | **81.7** | - | **87.8** | **87.1** | **87.1** | **86.9** | **86.8** |

**Weight Change Distribution.** We conduct an experiment on the DialogSum dataset using the Gemma-2b model to observe the impact of data samples on the weight change distribution across

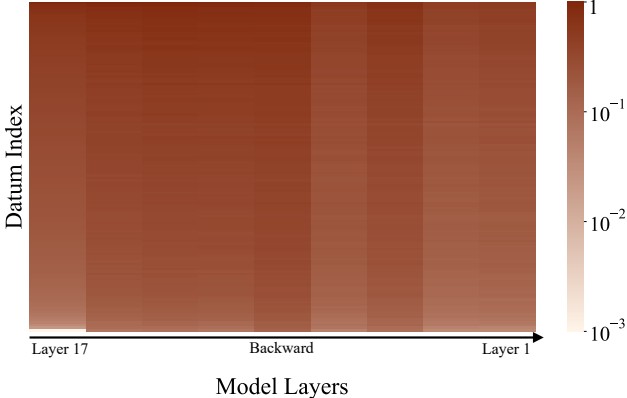

**Figure 5:** Samples that bring about large changes in the parameters of one layer can usually also significantly change the parameters of other layers.

different layers. The normalized weight changes are shown in Figure 5. We can observe that the weight change distributions on different layers are similar. Samples (i.e., important and hard samples) that bring significant weight changes to one layer can usually also significantly change the weights of other layers. This observation is consistent with the results of the gradient norm in Wang et al. (2024b). Following the open-sourced code in existing work (Paul et al., 2021; Lin et al., 2024), STAFF implements the effort score to measure how much effort the model makes to fit the data samples. Specifically, it calculates the L2 norm of the gradient matrix for each data sample and finally normalizes the effort scores on all data samples.

**Similarity Between Diverse Regions Estimated by Small and Target Models.** We use the Rand Index (RI) (Hubert & Arabie, 1985) to evaluate the similarity between the diverse data regions estimated by the small model and the target model. 1 indicates a perfect match, and 0 indicates a complete mismatch. We calculate the RI scores for different tasks and models, as shown in Table 18. We can observe that all RI scores are above 0.92 on various model families and tasks, which is close to a perfect match. The results indicate the consistency between the small model and the target model in partitioning diverse data distributions. To some extent, the data regions divided by the small model can represent the diverse distribution of the data for the target model. Therefore, STAFF can effectively utilize the data scores and divided data regions of the small model to allocate budgets and select data from different regions, ensuring data diversity while covering data samples important to the target model.

**Table 18:** RI scores on different models and tasks.

| Model | Random Index Score | | |
|---|---|---|---|
| | BioInstruct | DialogSum | WMT-19 |
| Gemma-7b VS 2b | 0.93 | 0.94 | 0.94 |
| Llama-2-13b VS 7b | 0.93 | 0.94 | 0.93 |
| Mistral-Nemo VS7B | 0.93 | 0.94 | 0.92 |

## A.5 DISCUSSION.

**Using on small models.** This paper introduces STAFF that utilizes a smaller model in the same family as the target LLM to select the coreset for task-specific fine-tuning of the target LLM. For the coreset selection on the models that are the smallest in the family (e.g., Llama-2-7b), using model pruning to build small models or selecting models from other families are potential solutions. Studying the impact of pruned models and models in another family on coreset selection and improving the selection method will be a future research direction.

**Impact of small model capability.** The capability of the small model significantly impacts the effectiveness of STAFF. A less capable small model cannot effectively learn knowledge from downstream tasks, thus failing to provide meaningful scores to guide data selection for the target LLM, leading to results close to the random selection. Miao et al. (2024) proposes to combine

multiple small models to achieve better performance in LLM speculative decoding, providing a potential direction for improving STAFF.

**Small models for coreset selection.** Some coreset selection methods for instruction fine-tuning also use small models to evaluate and select data. Du et al. (2023) focuses on using multiple models to evaluate data from the perspectives of quality, coverage, and necessity. Their evaluation models need to be pre-built and trained on human feedback datasets. Chen et al. (2023) encodes and clusters samples with K-mean and it then collects representative samples to construct the coreset. Different from these methods, STAFF does not require additional construction and training of new evaluation models. It extends the concept of speculative execution to coreset selection, using a small model to obtain a similar data score distribution as the target model, thus achieving better selection results and lower overhead than baseline methods at different pruning rates.

