# OpenReview forum: "STAFF: Speculative Coreset Selection for Task-Specific Fine-tuning"
_ICLR.cc/2025/Conference — ICLR 2025 Poster_

### Official Review · Reviewer_Mssy · 2024-10-30

**Soundness:** 3
**Presentation:** 3
**Contribution:** 3
**Rating:** 6
**Confidence:** 2

**Summary:**

The paper presents a novel method called STAFF for efficient and effective coreset selection in task-specific fine-tuning of large language models (LLMs). The paper claims to address two challenges from existing coreset selection methods: (1) Balancing data importance and diversity across pruning rates. (2) High overhead from the need to train the target LLM for several epochs to evaluate data scores and regions during selection.

The proposed STAFF method leverages similar ideas from speculative decoding to do speculative score calculation using small models and verification score from big models. A selection budget is calculated for each region based on the verification result, allocating more budget to regions deemed important to the target LLM while ensuring coverage of diverse data regions.

Experiment Results demonstrate that STAFF outperforms state-of-the-art coreset selection methods in both performance and efficiency across various pruning rates, LLMs, and downstream tasks

**Strengths:**

1. The experiment results look strong. The paper also includes detailed ablation studies.
2. The paper emphasizes reproducibility by providing access to the code and data used in the experiments.

**Weaknesses:**

1. The paper has limited incremental innovation on top of the work [Zhengetal.(2023)]"COVERAGE-CENTRIC CORESET SELECTION FOR HIGH PRUNING RATES". The [Zhengetal.(2023)] already did the research into the data importance and data diversity issue. This paper basically follows the similar way as [Zhengetal.(2023)] but with focus on the speculative implementation.
2. The paper doesn't have comprehensive related work study. For example, "Mods: Model-oriented data selection for instruction tuning" and
"Maybe only 0.5% data is needed: A preliminary exploration of low training data instruction tuning" also explored the small model for coreset selection. Given these two work already using small models for data selection. It might be helpful to discuss the difference of this paper with the above two papers.
3. The paper misses some theoretical analysis and proof for the equation (2)? For example, is that possible to define some theorem to get the connection of the equation (2) with loss ?

**Questions:**

How does the equation (2) come from?
Why not apply some normalization method for each weight change? For example, each weight difference normalized by the average of weight difference in each matrix or each layer.
Did you observe weight difference distribution across different layers?

---

> ### Author Response · Authors · 2024-11-21
> **Reply to Official Review by Reviewer Mssy [1/2]**
>
> Thank you for your valuable questions and suggestions. Our detailed responses are as follows.
>
> > **W1**. The paper has limited incremental innovation on top of the work [Zhengetal.(2023)]"COVERAGE-CENTRIC CORESET SELECTION FOR HIGH PRUNING RATES". The [Zhengetal.(2023)] already did the research into the data importance and data diversity issue. This paper basically follows the similar way as [Zhengetal.(2023)] but with focus on the speculative implementation.
>
> **R1**: STAFF and [Zhengetal.(2023)] (i.e., CCS) are different methods.
> CCS relies on fine-tuning the target model to calculate data scores and selects data from different regions with the same budget. This method works well on DL models but leads to huge selection overhead on LLMs with billions of parameters.
> In contrast, STAFF focuses on obtaining the data score distribution similar to that of the target model with less overhead and allocates more budget to data that is important to the target model, thereby effectively selecting data with lower overhead at different pruning rates.
>
> The major contribution is that STAFF innovatively extends the concept of Speculative Execution (which has been well-defined and widely applied in accelerating LLM inference) to coreset selection, **providing a new perspective for coreset selection work and the promotion and application of speculative execution concepts**.
> In large-scale experiments (totaling over 5,000 GPU hours), STAFF outperforms SOTA methods like CCS in terms of selection effectiveness at different pruning rates while significantly **reducing the selection overhead of CCS by up to 61.0%**.
>
> --------------
> > **W2**. The paper doesn't have comprehensive related work study. For example, "Mods: Model-oriented data selection for instruction tuning" and "Maybe only 0.5% data is needed: A preliminary exploration of low training data instruction tuning" also explored the small model for coreset selection. Given these two work already using small models for data selection. It might be helpful to discuss the difference of this paper with the above two papers.
>
> **R2**: Thank you for your valuable suggestion for improving our manuscript. We have supplemented these related papers in the revised version (see Line 119/1245).
> There are mainly two aspects of differences between STAFF and these works.
> 1) **Setting**. "Mods" and "Maybe" are designed for LLM instruction fine-tuning, which is a different fine-tuning setting from the task-specific fine-tuning that STAFF focuses on. [1]
> 2) **Method**. "Mods" focuses on using multiple models to evaluate data from the perspectives of quality, coverage, and necessity. These evaluation models need to be pre-built and trained on human feedback data. "Maybe" encodes and clusters samples with K-means and cosine similarity and it then collects representative samples to construct the coreset, which is similar to the baseline method `D2 pruning`.
> Different from these methods, STAFF extends the concept of Speculative Execution to coreset selection, using a small model to obtain a similar data score distribution as the target model, thus achieving better selection results and lower overhead than baseline methods (e.g., D2 pruning and CCS) at different pruning rates.
> In addition, STAFF does not require additional construction and training of new evaluation models.
>
> [1] A Survey on Data Selection for Language Models. TMLR 2024

---

> > ### Comment · Reviewer_Mssy · 2024-11-26
> >
> > I would like to express my thanks to the authors for providing detailed responses to my questions. After carefully reviewing their rebuttals to other reviewers' comments, I updated my initial rating for this paper.

---

> > > ### Author Response · Authors · 2024-11-26
> > >
> > > Thank you for your positive feedback on our responses and the revised manuscript.
> > >
> > > We remain open and willing to address any additional questions or concerns you may have.

---

> ### Author Response · Authors · 2024-11-21
> **Reply to Official Review by Reviewer Mssy [2/2]**
>
> > **W3**.The paper misses some theoretical analysis and proof for the equation (2)? For example, is that possible to define some theorem to get the connection of the equation (2) with loss?
> Q1: How does the equation (2) come from? Why not apply some normalization method for each weight change? For example, each weight difference normalized by the average of weight difference in each matrix or each layer. Did you observe weight difference distribution across different layers?
>
> **R3**: Thank you for your question.
> 1. As shown in Line 249, Eq (2) is from prior works [2][3] and its effectiveness has been demonstrated in these works. Following your suggestion, we have supplemented the theoretical analysis of Eq (2) in the revised version (see Line 848).
> 2. The current implementation of Eq (2) in STAFF follows the code of prior work[2][3], where it calculates the L2 norm of the gradient matrix for each data sample to obtain the corresponding effort score, and finally normalizes the scores on all data samples in the calculation to avoid the impact of outliers on coreset selection.
> 3. Following your suggestion, we have conducted an experiment to observe the distribution of weight changes across different layers.
> We have observed that the weight change distributions on different layers are similar, which is consistent with the observations of prior work[4]. Specifically, when learning a sample brings a large weight change to a certain layer (e.g., Layer 17), it usually also leads to large weight changes on other layers (e.g., Layer 1). We have added the corresponding analysis in the revised version (see Line 1187).
>
>
> [2] Deep Learning on a Data Diet: Finding Important Examples Early in Training. NeurIPs 2021
>
> [3] Data-efficient Fine-tuning for LLM-based Recommendation. SIGIR 2024
>
> [4] Efficient Backpropagation with Variance-Controlled Adaptive Sampling. ICLR 2024

---

### Official Review · Reviewer_PBZt · 2024-10-30

**Soundness:** 4
**Presentation:** 3
**Contribution:** 4
**Rating:** 8
**Confidence:** 5

**Summary:**

This paper presents an efficient coreset selection algorithm for LLMs which is performant at high pruning rates. The key ideas are to 1) use a smaller LLM from the same model family as the larger LLM to compute an example score and verify that score using the larger LLM, and 2) allocate selection budget to regions which are more important for the target LLM.  Results on Gemma-7b, Llama-2-13b and Mistral-Nemo show that the method can outperform other baselines at both high and low pruning rates (up to 54.3%) with a lower time overhead (up to 70%).

**Strengths:**

* Presents an approach for core set selection which is both performant and efficient compared to existing baselines
* Demonstrates improvements in performance and time-overhead for three LLMs across a range of pruning rates
* Presents ablations showing the role of each component (verification step, smaller model, smaller model from a different family)

**Weaknesses:**

* Some parts of the paper are hard to follow. See questions below.
Note: The authors have addressed the clarity issues in their updated version.

**Questions:**

Questions
* It is hard to decipher whether the target LLM (\theta_t) used in verification (Step 10 in Algorithm 1) is a fine-tuned model. Algorithm 1 suggests that it is not fine-tuned but it would be good if this can be clarified.
* L184 says: "without extensive fine-tuning of the target LLM." Does this imply that some fine-tuning was done? Please clarify.
* L247 says: "After selecting and fine-tuning θs, we introduce the effort score ". It looks like the effort score was computed after fine-tuning the smaller LLM and prior to data selection. Please clarify.
* Table 1: It would be useful to report the performance of the fine-tuned small LLM used in coreset selection.

Typos:
* L045: are difficult -> have difficulty
* L297: 'use' -> 'uses'
* L731: 'baslines' -> 'baselines'

---

> ### Author Response · Authors · 2024-11-21
> **Reply to Official Review by Reviewer PBZt**
>
> Thank you for your careful review of our manuscript and your valuable suggestions. Our detailed responses are as follows.
>
> > **Q1**. It is hard to decipher whether the target LLM (\theta_t) used in verification (Step 10 in Algorithm 1) is a fine-tuned model. Algorithm 1 suggests that it is not fine-tuned but it would be good if this can be clarified.
>
> **R1**: Thanks for your question.  $\theta_t$ is the target model without fine-tuning. STAFF fine-tunes the small model  $\theta_s$ in the same family as $\theta_t$ to evaluate the data scores and then uses $\theta_t$ for validation and selection.
> Compared with baselines (e.g., CCS) that directly fine-tune $\theta_t$ to evaluate data scores, STAFF can perform effective coreset selection with lower overhead (Table 2).
> We have clarified in the revised version (see Line 208/273).
>
> -------------
> > **Q2**. L184 says: "without extensive fine-tuning of the target LLM." Does this imply that some fine-tuning was done? Please clarify.
>
> **R2**: Thank you for pointing out the ambiguous description. STAFF does not need fine-tuning on the target model during coreset selection (which is heavy), thereby reducing the overhead of coreset selection. We have removed the ambiguous `extensive` in the revised version (see Line 184).
>
>
> -------------
> > **Q3**. L247 says: "After selecting and fine-tuning θs, we introduce the effort score ". It looks like the effort score was computed after fine-tuning the smaller LLM and prior to data selection. Please clarify.
>
> **R3**: Thank you for pointing out the ambiguous description. After selecting a small model $\theta_s$ from the same family as $\theta_t$ and fine-tuning it on the dataset, we use the effort score (i.e., the effort of the model $\theta_s$ in learning each data sample) to evaluate the speculative score of the data, which will be used for subsequent verification and coreset selection.
> Here in Line 247, "selecting" refers to selecting the small model $\theta_s$ from the same family as $\theta_t$, not selecting data samples. We have removed the ambiguous `selecting` in the revised version (See Line 247).
>
>
> -------------
> > **Q4**.Table 1: It would be useful to report the performance of the fine-tuned small LLM used in coreset selection.
>
> **R4**: Thanks for your valuable suggestion. We have reported the performance of the target models and corresponding small models (w/o fine-tuning) in Table 4. Following your suggestion, we have updated the performance of the fine-tuned small and large models on the complete dataset in this table. (see Line 814)
> The performance of these models has been significantly improved after fine-tuning. For example, the ROUGE-L of Gemma-7b and 2b models on the WMT-19 dataset are improved from 0.2 and 0.3 to 62.2 and 53.4, respectively.
> Leveraging these fine-tuned small models, STAFF effectively and efficiently evaluates data scores and selects coresets for the target models.
>
>
>
> -------------
> > **Q5**.Typos:
> L045: are difficult -> have difficulty
> L297: 'use' -> 'uses'
> L731: 'baslines' -> 'baselines'
>
> **R5**: Thank you for your suggestion for improving our manuscript. We have fixed these typos in the revised version and thoroughly checked the entire manuscript to avoid similar problems (see Line 45/297/785).

---

> > ### Comment · Reviewer_PBZt · 2024-11-21
> >
> > Thanks for answering my questions and updating the paper.

---

> > > ### Author Response · Authors · 2024-11-24
> > >
> > > Thank you for taking the time and effort to review our responses and the revised manuscript.
> > > We remain open and willing to address any further questions or concerns you may have.

---

### Official Review · Reviewer_nbb8 · 2024-10-31

**Soundness:** 4
**Presentation:** 4
**Contribution:** 3
**Rating:** 6
**Confidence:** 4

**Summary:**

The paper introduces a novel method for improving the efficiency of large language model (LLM) fine-tuning by reducing the computational resources and time required. The authors propose STAFF, a speculative coreset selection method that leverages a small model from the same family as the target LLM to estimate the importance of data samples. This approach allows for the identification of important data regions while maintaining diversity, leading to a more efficient selection process with lower overhead. The paper evaluates STAFF on three different LLMs and three downstream tasks, demonstrating that it can improve the performance of state-of-the-art methods by up to 54.3% and reduce selection overhead by up to 70.5% at various pruning rates.

**Strengths:**

The paper presents a creative solution to the problem of resource-intensive fine-tuning of LLMs by introducin a speculative coreset selection method coreset.

The paper demonstrates that STAFF can reduce the selection overhead by up to 70.5% compared to other methods, which is a substantial improvement for practical applications where time and computational resources are critical.

The proposed method can selected the important samples (and also the important sample for the target LLM) while keeping the diversity.

**Weaknesses:**

The effectiveness of STAFF relies heavily on the small model's capability to estimate the importance of data samples accurately. If the small model is not sufficiently capable, the coreset selection may not be effective.

For the gradient used in paper, the gradient of small LLM is from a LLM that has been finetuned. However, the gradient of target LLM is from a LLM that is not finetuned. Those two gradient might not be comparable for the calculation of Verification score. Maybe a sample that is hard for a small LLM is easy for a larger LLM?

**Questions:**

Is there result for more tasks?

---

> ### Author Response · Authors · 2024-11-21
> **Reply to Official Review by Reviewer nbb8**
>
> Thank you for your valuable questions and suggestions. Our detailed responses are as follows.
>
> > **W1**. The effectiveness of STAFF relies heavily on the small model's capability to estimate the importance of data samples accurately. If the small model is not sufficiently capable, the coreset selection may not be effective.
>
> **R1**:  Thank you for your question.
> The effectiveness of methods based on speculative execution (including STAFF and existing LLM speculative decoding methods[1][2]) is indeed affected by the capability of the small model used to complete the speculative task.
> The difference between the knowledge distribution of the small model and the target model can make STAFF fail to effectively obtain a similar score distribution to the target model, resulting in reduced selection effectiveness.
> We have conducted an experiment to discuss the impact of using a model with different score distributions in the ablation study (see Table 3 `other small model` and Figure 4).
> We have further discussed the impact of the small model's capability on STAFF and potential enhancement in the Appendix (see Line 1239).
> We recommend users use small models from the same family as the target model, as they have similar pre-trained knowledge as the large model, which helps to achieve better coreset selection performance.
>
> [1] Fast inference from transformers via speculative decoding. ICML 2023
>
> [2] Speculative decoding: Exploiting speculative execution for accelerating seq2seq generation. EMNLP 2023
>
> -----------------------
> > **W2**. For the gradient used in paper, the gradient of small LLM is from a LLM that has been fine-tuned. However, the gradient of target LLM is from a LLM that is not fine-tuned. Those two gradient might not be comparable for the calculation of Verification score. Maybe a sample that is hard for a small LLM is easy for a larger LLM?
>
> **R2**: Thank you for your question. Prior work and our experimental results in Table 12 (see Line 1039) show that fine-tuning the model helps evaluate the data and perform coreset selection more effectively.
> The situation you mentioned where a set of data has different importance for large and small models does exist. To solve such inconsistency, in the Verification & Selection stage (Section 3.2), STAFF dynamically modifies the selection budget for different data regions based on the difference in scores between the small and large models on the same regions and allocates more budget to regions that are more important to the large model.
> In your example, when the samples are difficult for the small model and easy for the target model, it will result in a high $S^s_d$, low $S^t_d$, low $V_i$, and low selection budget $m_b$ (according to Eq(4)). Therefore, STAFF will select these samples in small quantities and allocate more budget to other important/difficult samples for the target model.
>
> [3] Deep Learning on a Data Diet: Finding Important Examples Early in Training. NeurIPs 2021
>
> -----------------------
> > **Q1**.Is there result for more tasks?
>
> **R3**: Thanks for your suggestion. Following your suggestion, we have supplemented experiments on the paraphrasing task (`NoaiGPT/ltgen-wiki-paraphrased-Humanized-19999` in HuggingFace) to show the effectiveness of STAFF on more tasks.
> This dataset includes 19,999 pairs of Wikipedia paragraphs and corresponding more humane and natural paraphrases.
> We have supplemented the results in the revised version (see Line 1160). Existing results are shown as follows.
> We can observe that in the paraphrasing task, the coreset selection effect of STAFF still outperforms the baselines at different pruning rates, and can improve the baseline results by up to 65.72% at p=90%.
> Note that, at a low pruning rate of p=20%, the coreset selected by STAFF achieves better fine-tuning effects than the complete dataset.
> Such an observation is consistent with the results in Table 1 of our manuscript.
> It further illustrates the effectiveness of STAFF in selecting coresets and improving data efficiency for LLMs on various tasks.
> In addition, the selection overhead of STAFF is only 2.4 hours (p=90%), which is 58.7% to 72.3% shorter than that of the baseline methods.
>
> |   Model  |   Method   | ROUGE-L |     |     |      |      |      |
> |:--------:|:----------:|:-------:|:---:|:---:|:----:|:----:|:----:|
> |          |            |  0%  |  20% |  50% |  70% |  80% |  90% |
> | Gemma-7b |   Random   | 82.5 | 81.9 | 81.3 | 80.7 | 80.9 | 80.6 |
> |          |    GraNd   |   -  | 81.9 | 77.0 | 58.6 | 49.7 | 49.3 |
> |          |    EL2N    |   -  | 80.9 | 69.2 | 63.7 | 64.2 | 64.6 |
> |          |     CCS    |   -  | 82.2 | 81.6 | 80.7 | 81.1 | 80.5 |
> |          | D2 Pruning |   -  | 81.6 | 70.7 | 64.7 | 63.2 | 64.5 |
> |          |    STAFF    |   -  | 83.0 | 81.9 | 81.9 | 81.8 | 81.7 |

---

> > ### Comment · Reviewer_nbb8 · 2024-11-26
> > **Thanks for your reponse**
> >
> > I have read the author response. Thanks for the effort of the authors in this period. Most of my concerns are addressed.

---

> > > ### Author Response · Authors · 2024-11-26
> > >
> > > Thank you for providing valuable insights and taking the time to review our responses and the revised manuscript.
> > >
> > > We remain open and willing to address any further questions or concerns you may have.

---

### Official Review · Reviewer_iT4U · 2024-11-08

**Soundness:** 3
**Presentation:** 3
**Contribution:** 3
**Rating:** 6
**Confidence:** 4

**Summary:**

This paper proposes an improved coreset selection method for task-specific fine-tuning of LLMs which consists of two stages: speculative score calculation by leveraging a small LLM of same structure, and LLM verification and selection which dynamically allocate selection budgets for easy and difficult regions of data samples. The authors demonstrate the effectiveness and efficiency of their proposed method by comparing to five SOTA selection methods, including random, GraNd, EL2N, CCS, and D2 Pruning, on three different tasks, including BioInstruct (QA), DialogSum (summarization), and WMT-19 (translation), for three different LLMs, including Gemma-7b, Llama-2-13b, and Mistral-Nemo-Instruct-2407. The results show the proposed method outperforms other baselines by a significant margin.

**Strengths:**

1. In general this paper is well-written and the authors demonstrate their methodology in a clear way. The experimental part of this paper is also persuasive with broad ranges of tasks, models, and comparison with different baselines.

2. Although speculative decoding is well-defined and applied for optimizing LLM decoding, incorporating this approach for coreset selection can not only broaden the scope of application, but also fosters creativity and novel insights.

3. An effective and efficient coreset selection method is indeed important for real-world LLM applications. If the proposed approach can be generalized for a broad range of tasks and models, it could be of great significant for building new applications based on LLMs.

**Weaknesses:**

1. Although the authors show extensive experimental results, it will be of great significant to show if the proposed method can be scalable for different model sizes, especially for bigger models.

2. Some parts of this work should be clarified. Please refer to my questions.

**Questions:**

1. I'm a bit concerned on maintaining the diversity of the coreset data. Basically the authors utilize the scores estimated from a smaller model to split the dataset into different regions which represent the diverse distribution of the data. The authors only update their estimation of importance during the verification stage without any modification of the regions. Could the authors quantify the difference of diversity estimated by the small and the target models?

2. It seems their is no ablation study estimation better combination of small and target models, e.g. how small the small model could be to make it more efficient?

---

> ### Author Response · Authors · 2024-11-21
> **Reply to Official Review by Reviewer iT4U [1/2]**
>
> Thank you for your valuable insights and suggestions. Our detailed responses are as follows.
>
> > **W1**. Although the authors show extensive experimental results, it will be of great significant to show if the proposed method can be scalable for different model sizes, especially for bigger models.
>
> **R1**: Thank you for your valuable insights. Our work shares the foundation with existing speculative decoding work. Existing speculative decoding work [1] has demonstrated that the small model (e.g.,  LLaMA-68M) can guide and accelerate the inference of a much larger model in the same family (LLaMA-65B). STAFF, which is also built on the concept of speculative execution, can theoretically use a small model to guide data selection for a larger model (e.g., 32B).
>
> Following your suggestion, we have conducted experiments on models in the Qwen2.5 family to verify this. We use the 32B model as the target model and separately use 3B/7B/14B models as the speculative model.
> The results are shown in the following table (DialogSum dataset, ROUGE-L).
> We can observe that for the target model with over 30B parameters, STAFF can still use the small model with different sizes to effectively guide coreset selection, which is consistent with the above theoretical analysis.
> Even using a 3B model with only 1/10 the number of parameters of the target model, STAFF can still achieve better selection results than the baselines.
>
> Additionally, STAFF can efficiently select data for the large target model.
> Benefiting from the low fine-tuning overhead of the small models, when p=90%, STAFF separately requires 3.0/3.1/5.3 hours to perform coreset selection using 3B/7B/14B models, reducing the selection overhead by up to 89.7% compared to baseline methods (usually needing over 26 hours fine-tuning the model to evaluate data scores).
> Note that using a larger speculative model has the potential to obtain data score distribution more similar to the target model, leading to better coreset selection results.
> However, it will also bring greater selection overhead.
> For example, the selection overhead of the 14B version is 1.77 times that of the 3B version.
> Considering the limited performance improvement brought by such a significant increase in selection overhead, we recommend users choose the smallest possible officially released model in the model family (e.g., those with a size of less than 7B) as the speculative model.
> We have supplemented the results in the revised version (see Line 1120).
>
> |Model|Method|ROUGE-L||||||
> |-|-|-|-|-|-|-|-|
> |||0%|20%|50%|70%|80%|90%|
> |Qwen2.5-32B|Random|50.2|49.5|48.6|48.1|47.5|47.1|
> ||GraNd|-|50.5|48.6|47.5|46.9|45.6|
> ||EL2N|-|50.5|48.9|48.2|47.6|46.3|
> ||CCS|-|50.2|49.3|49.1|49.0|48.6|
> ||D2Pruning|-|49.7|49.7|48.9|47.2|46.8|
> ||STAFF-3b|-|50.5|50.0|49.7|49.0|48.9|
> ||STAFF-7b|-|50.5|50.2|49.6|49.2|49.1|
> ||STAFF-14b|-|51.1|50.1|49.6|49.3|49.1|
>
> [1] Specinfer: Accelerating large language model serving with tree-based speculative inference and verification. ASPLOS 2024.
>
>
> -----------------------
> > **Q1**. I'm a bit concerned on maintaining the diversity of the coreset data. Basically the authors utilize the scores estimated from a smaller model to split the dataset into different regions which represent the diverse distribution of the data. The authors only update their estimation of importance during the verification stage without any modification of the regions. Could the authors quantify the difference of diversity estimated by the small and the target models?
>
> **R2**: Thank you for your question. To verify whether the data regions divided by the small model can represent the diverse distribution of data on the target model, we use the Rand Index (RI) [2] to quantify and evaluate the similarity between the diverse data regions (clusters) estimated by the small model and the target model, where 1 means a perfect match, and 0 means a complete mismatch. The RI scores are shown as follows.
>
> |Model| BioInstruct | DialogSum | WMT-19 |
> |-|-|-|-|
> |Gemma-7b VS 2b | 0.93 |0.94|0.94|
> |Llama-2-13b VS 7b |0.93|0.94|0.93|
> |Mistral-Nemo VS7B | 0.93|0.94|0.92|
>
> We can observe that all RI scores are above 0.92 on various model families and tasks, which is close to a perfect match.
> The results indicate the similarity between the small model and the target model in partitioning diverse data distributions.
> There are still differences in the distribution between target and small models, therefore, in the verification stage, STAFF adjusts the selection budget based on the difference in scores between the small and target models on different data regions.
> As a result, it can allocate more budgets for difficult/important samples for the large model at low pruning rates, while also covering diverse data regions at high pruning rates, ultimately achieving efficient and effective coreset selection.
> We have supplemented the discussion in the revised version (see Line 1212).
>
>
> [2] Comparing partitions. Journal of classification 1985

---

> ### Author Response · Authors · 2024-11-21
> **Reply to Official Review by Reviewer iT4U [2/2]**
>
> > **Q2**.It seems their is no ablation study estimation better combination of small and target models, e.g. how small the small model could be to make it more efficient?
>
> **R3**: Thank you for your valuable suggestion.
> This is a similar question to W1, that is, how to combine large and small models in STAFF (larger target models or smaller speculative models) to select data efficiently.
> The efficiency of coreset selection is related to the time overhead of fine-tuning the small model and evaluating the speculative score.
> Compared to the target model, the smaller the speculative model is, and the faster it fine-tunes, the higher the efficiency of STAFF in selecting coresets.
> For example, in R1, STAFF uses the Qwen-2.5-3b model to select the coreset for the 32b model, which can reduce the time cost by nearly 90% compared to the baselines. The Qwen-2.5-14b version can reduce the time overhead by about 80%.
> Using Llama-2-7b to select data for the 13b model may only reduce the overhead by 18.2% compared to the baseline methods.
>
> Following your suggestion, we have added an ablation study to study the impact of smaller models with different sizes (see Line 1089).
> We use two pruned versions of the Llama-2-7b published on HuggingFace, with 50% and 70% of parameters pruned by SparseGPT[3]. We compare the coreset selection results using these models with the original Llama-2-7b (i.e., selecting coresets for Llama-2-13b on the DialogSum dataset), and the results are as the following table.
> We can see that the performance degradation of the model with a 70% pruning rate is more significant than that of the model with a 50% pruning rate.
> The selection time overhead of the 50% and 70% pruned models is reduced by 2.4% and 3.8% compared to the original Llama-2-7b model, respectively (which may be affected by the pruning method), which is still consistent with our theoretical analysis.
> That is, a faster fine-tuning of the smaller model (compared to the target model)  can improve the efficiency of the coreset selection in STAFF.
>
>
> |        Method       | ROUGE-L |      |      |      |      | BLEU-4 |      |      |      |      |
> |:-------------------:|:-------:|:----:|:----:|:----:|:----:|:------:|:----:|:----:|:----:|:----:|
> |                     |   20%   |  50% |  70% |  80% |  90% |   20%  |  50% |  70% |  80% |  90% |
> |       Llama-7b      |   50.0  | 49.1 | 49.0 | 48.4 | 48.2 |  52.8  | 51.9 | 51.7 | 51.0 | 50.1 |
> | Llama-7b-Pruned 50% |   49.9  | 49.1 | 49.0 | 48.5 | 48.1 |  52.6  | 51.6 | 51.6 | 50.9 | 50.2 |
> | Llama-7b-Pruned 70% |   49.6  | 49.1 | 48.9 | 48.2 | 47.7 |  52.4  | 51.3 | 51.3 | 50.9 | 50.1 |
>
> In addition, the similarity in knowledge distribution and function between the small model and the target model is the key determinant of STAFF's selection effectiveness.
> Smaller models could lead to faster execution speed, but they could result in degraded selection performance due to differences in data score distributions compared to the target model. There is a trade-off between data selection effectiveness and efficiency.
> We currently recommend that users directly use officially released pre-trained small models from the same family to calculate speculative scores.
>
> Existing work lacks a comprehensive and effective metric to evaluate the similarity of functions and performance between models.
> For example, [4] proposed a series of metrics to evaluate the representational (functional) similarity between LLMs, but they observed that the evaluation results of different metrics varied greatly and were difficult to interpret.
> Building a framework to evaluate the similarity between large and small models is a potential enhancement for STAFF to select effective and efficient small models in the future.
>
> [3] Sparsegpt: Massive language models can be accurately pruned in one-shot. ICML 2023
>
> [4] Towards Measuring Representational Similarity of Large Language Models. NeurIPS workshop 2023

---

### Author Response · Authors · 2024-11-21
**General Response**

We thank the reviewers for their thorough work in evaluating our manuscript, and their thoughtful comments that contribute significantly to its improvement. Following reviewers‘ suggestions, we have clarified and supplemented the manuscripts in the following aspects. (All revisions have also been updated in the revised manuscripts)
1.  We have added more experiment results and details, including the coreset selection for a larger target model (i.e., 32B parameters), coreset selection in other tasks (i.e., a human label dataset of the paraphrasing task), and the quantitative measurement for the similarity of diverse data regions estimated by small models and target models. (e.g., Line 1119/1149/1183)
2. We have clarified the method details and theoretical analysis in the manuscript and fixed some ambiguous descriptions and typos. (e.g., Line 208/273/848/1173)
3. We have added a discussion of related work, highlighting the novelty and contributions of our manuscript, namely extending the concept of Speculative Execution (which has been applied in accelerating LLM decoding) to LLM coreset selection, providing a new perspective for coreset selection and the promotion and application of speculative execution concepts. (e.g., Line 119/1232)

---

### Author Response · Authors · 2024-11-24
**An earnest request to check the responses and confirm if there are any further questions**

Dear reviewers,

Thank you for your time and constructive feedback on our manuscript.
We have completed the supplemental experiments and updated the experimental results and analysis in responses and the revised manuscript (Line 1120/1160).

Would you mind checking our responses and confirming whether you have any further questions?

We remain open and willing to address any further questions or concerns you may have.

Thanks again for your thoughtful comments and best regards.

---

### Meta-Review · Area_Chair_cN2Z · 2024-12-07

**Metareview:**

The authors propose STAFF, a speculative coreset selection method that leverages a small model from the same family as the target LLM to estimate the importance of data samples. This approach enables the identification of significant data regions while preserving diversity, resulting in a more efficient selection process with reduced overhead. The authors demonstrate the effectiveness and efficiency of their proposed method by comparing it to five selection methods across three different tasks and three different LLMs. The results show that the proposed method significantly outperforms other baselines. A reviewer raises concerns about the novelty of the work and mentions some similar prior research. In their rebuttal, the authors address the reviewer's concerns.

**Additional Comments On Reviewer Discussion:**

Reviewer nbb8 points out that the success of the approach heavily relies on the small model. The authors’ rebuttal addresses this concern. Reviewer Mssy raises concerns that the paper offers limited incremental innovation. The authors’ rebuttal addresses this issue.

---

### Decision · Program_Chairs · 2025-01-22

Accept (Poster)